# High-resolution global map of closed-canopy coconut palm

Adrià Descals[1], Serge Wich[2], Zoltan Szantoi[3,4], Matthew J. Struebig[5], Rona Dennis[6], Zoe Hatton[2], Thina Ariffin[6], Nabillah Unus[6], David L.A. Gaveau[7,8], and Erik Meijaard[6]

[1]CREAF, Cerdanyola del Vallès, Barcelona 08193, Spain
[2]School of Biological and Environmental Sciences, Liverpool John Moores University, Liverpool L3 3AF, United Kingdom
[3]Science, Applications & Climate Department, European Space Agency, Frascati 00044, Italy
[4]Stellenbosch University, Stellenbosch 7602, South Africa
[5]Durrell Institute of Conservation and Ecology, University of Kent, Canterbury, England, United Kingdom
[6]Borneo Futures, Bandar Seri Begawan, Brunei Darussalam
[7]TheTreeMap; Bagadou Bas, Martel 46600, France
[8]Visiting Professor, Jeffrey Sachs Center on Sustainable Development, Sunway University, 5, Jalan Universiti, Bandar Sunway, 47500 Petaling Jaya, Selangor

*Correspondence to*: Adrià Descals (a.descals@creaf.uab.cat)

**Abstract.** Demand for coconut is expected to rise, but the global distribution of coconut palm is understudied, which hinders the discussion of its impacts. Here, we produced the first 20-meter global coconut palm layer using a U-Net model that was trained on annual Sentinel-1 and Sentinel-2 composites for the year 2020. The overall accuracy was $99.04 \pm 0.21$ %, which was significantly higher than the no-information rate. The producer's accuracy for coconut palm was $71.51 \pm 23.11$ % when only closed-canopy coconut palm was considered in the validation, but decreased to $11.30 \pm 2.33$ % when sparse and dense 20 open-canopy coconut palm was also taken into account. This indicates that sparse and dense open-canopy coconut palm remains difficult to map with accuracy. We report a global coconut palm area of $12.66 \pm 3.96$ x $10^6$ ha for dense open- and closed-canopy coconut palm, but the estimate is three times larger ($38.93 \pm 7.89$ x $10^6$ ha) when sparse coconut palm is included in the area estimation. The large area of sparse coconut palm is important as it indicates that production increases can likely be achieved on the existing lands allocated to coconut. The Philippines, Indonesia, and India account for most of the global 25 coconut palm area, representing approximately 82 % of the total mapped area. Our study provides the high-resolution, quantitative, and precise data necessary for assessing the relationships between coconut production and the synergies and trade-offs between various sustainable development goal indicators. The global coconut palm layer is available at https://doi.org/10.5281/zenodo.8128183 (Descals, 2022).

## 1 Introduction

Coconut (Cocos nucifera L.) is a palm species native to tropical islands in the western Pacific but also grows in other tropical areas (Gunn et al., 2011). Climate is an important determinant of coconut palm growth and yield (Peiris and Thattil, 1998). Climate factors such as temperature and relative humidity have been used in descriptive models for predicting coconut yield up to four years in advance (Kumar et al., 2009a). Weather data explained past trends in coconut production (Kumar et al., 2009b), and potential changes in the coconut palm distribution area expected due to climate change in India (Hebbar et al., 2022). Coconut palms produce about 1.7% of the global volume of vegetable oils as well as copra, coconut water, and coconut milk. Coconut palm is generally overlooked in discussions about crop impacts and not many see this palm as a threat to biodiversity. However, a recent study identified coconut palms as a potential threat to tropical species, many of which are highly threatened and restricted to tropical islands where coconut palm is extensively grown (Meijaard et al., 2020). In some of these islands, coconut palm is considered an invasive species that drives near-complete ecosystem state change when it becomes dominant (Young et al., 2017).

Despite the potential impacts, the coconut palm distribution is poorly documented except for national-level statistics on estimated harvest areas (FAO, 2022), local-level crop mapping studies (e.g., Palaniswami et al., 2006), and global coarse-resolution modelling (Yu et al., 2020). This may be because coconut palm is mostly grown in smallholdings under 4 ha (Omont, 2001) and is often intercropped, making its mapping difficult. A high-resolution global map of the coconut palm distribution can be used in geospatial analysis to assess environmental impacts and, thus, inform policy (e.g., estimate the extent of coconut plantations in areas of high biodiversity and assess the subsequent impact on biodiversity indices). Research is therefore needed to map the extent of coconut palm on a global scale, especially using high-spatial resolution satellite data.

Sub-meter satellite data and aerial images have been used for detecting individual coconut palms (Zheng et al., 2023; Freudenberg et al., 2019; Zheng et al., 2021), delineating coconut palm canopy (De Souza and Falcão, 2020; Vermote et al., 2020), and coconut palm detection in the context of land cover classification (Burnett et al., 2019). These studies used various methodologies, including threshold-based classification, random forest using feature extraction, and more advanced techniques such as object detection and semantic segmentation using deep learning. Similar efforts have been made to map coconut palm using decametric-scale satellites such as Sentinel-1, Sentinel-2, or Landsat-7 (Lang et al., 2021; Jenifer and Natarajan, 2021; Palaniswami et al., 2006). Another study detected individual coconut palms using airborne laser scanning (Mohan et al., 2019). Despite previous efforts to map coconut palm, these studies have focused on the local and regional scale, and a global coconut palm map has not been produced yet at a high spatial resolution. Moreover, it is still unclear how well satellite remote sensing can differentiate between coconut palm and other palm species, in particular oil palm (Gibril et al., 2017). The confusion between coconut palm and oil palm explains the potential commission errors in previous oil palm datasets (Descals et al., 2021; Danylo et al., 2021; Gaveau et al., 2022).

This study aims to produce the first global coconut palm map at high spatial resolution (20 meters) and estimate the global coconut palm area using satellite remote sensing. To achieve this aim, we first identified potential areas where climate was favourable for coconut palm growth. We then used a semantic segmentation model to classify Sentinel-1 and Sentinel-2 annual composites for 2020. Finally, we employed a sampling-based approach to validate the results.

## 2 Methods

### 2.1 Overview

To map coconut palms globally, we first conducted a bioclimatic analysis to determine regions in the world where coconut can potentially grow. The bioclimatic analysis used climate variables and terrain slope to produce a map of the potential coconut palm distribution. The regions identified in the bioclimatic analysis served as the focus of our mapping efforts. The mapping of coconut palm consisted of a supervised classification of Sentinel-1 and Sentinel-2 data. Specifically, we selected

bands VV and VH from Sentinel-1 and band 11 from Sentinel-2 after evaluating their backscatter and spectral separability for different tree plantations. The selected bands (VV, VH, and band 11) were aggregated into annual composites, which were then used as input in the classification model. The classification model was a U-Net that predicted two classes: class 'coconut' and class 'other'. The model was deployed within the regions identified in the potential coconut palm distribution. To validate the resulting classification layer, we used a sampling-based approach with 10,200 reference points. Lastly, due to data

limitations in certain areas, such as the Pacific, we conducted a sampling-based estimation of coconut palm area in small tropical islands using sub-meter resolution satellite images.

### 2.2 Bioclimatic analysis for mapping the potential distribution of coconut palm

We used a bioclimatic analysis to determine the potential coconut-producing regions and, subsequently, constrain the spatial extent of the classification of satellite data. To achieve this, we first conducted a literature search to identify regions known

for coconut palm cultivation. Additionally, we used the SPAM2010 (Spatial Production Allocation Model) dataset (Yu et al., 2020), which depicts the global occurrence of coconut production across a 5-arcmin grid (Fig. A1). Once the coconut-producing regions were identified, we visualized sub-meter resolution satellite data shown in Google Earth and collected points in locations where coconut palms were present (Fig. 1a). Three interpreters visualized sub-meter resolution and collected at least five points in each SPAM grid cell. Coconut palm can be distinguished from other palm species in sub-meter satellite

images (Fig. 2). If available, the interpreters visualized images from Google Street View to verify the presence of coconut palms.

Once all coconut-producing regions were sampled, we extracted the values from a terrain slope layer and from the WorldClim V1 Bioclim (Hijmans et al., 2005) at the collected points. The terrain slope was derived from the Shuttle Radar Topography

Mission (SRTM) digital elevation dataset (Jarvis et al., 2008). WorldClim V1 Bioclim consists of 19 bioclimatic variables derived from monthly temperature and precipitation. Given that the variables WorldClim were obtained from the same time series, we used the variance inflation factor (VIF) to address collinearity issues. The VIF determines if a set of variables is strongly correlated with each other. A VIF value higher than 5 indicates a high multicollinearity. We removed variables that presented a VIF higher than 5, which resulted in a subset of 8 WorldClim variables and terrain slope (Table A1). We used only the subset of 9 variables in the estimation of the potential coconut palm distribution. The values in the 9 variables outlined the range of bioclimatic values for coconut palm, and we used this range (minimum and maximum) to generate the potential coconut palm distribution map; a pixel in the WorldClim dataset was considered suitable for coconut palm growth if the 9 selected variables fell within the bioclimatic range.

## 2.3 Sentinel-1 and Sentinel-2 compositing

Sentinel-1 and Sentinel-2 annual composites for the year 2020 were the input data for the classification model. Sentinel-1 consists of two SAR satellites with a 6-day revisit time (Torres et al., 2012). We used the polarization bands VV and VH, and the median was computed for all available observations in the ascending and descending scenes separately. The annual composite of Sentinel-1 was the mean of the two orbit composites for 2020. Sentinel-2 consists of two optical satellites that provide images at a revisit time of 5 days. We used the Sentinel-2 level-2A product, which provides terrain-corrected top-of-canopy reflectance. Non-valid observations were masked using the Scene Classification Layer, which is produced by the ATCOR algorithm for the level-2A product (Drusch et al., 2012). The Sentinel-2 annual composites were generated using the median of all available valid observations for 2020. The compositing for Sentinel-1 and Sentinel-2 was identical to that of the global oil palm layer described in Descals et al. (2021), with the exception that the global oil palm layer was created with images from the second half of 2019 rather than the entire 2020. Coconut species is an evergreen plant and its canopy does not show substantial seasonal changes that can be captured in Sentinel-1 and Sentinel-2. The annual compositing used in this study may not be effective for mapping crops and vegetation that present a distinctive land surface phenology, which can provide key information for successfully mapping their extent (Son et al., 2013).

## 2.4 Feature selection

The coconut palm classification follows a methodology similar to that used for the global oil palm layer (Descals et al., 2021). The classification comprised a semantic segmentation model that used three input layers. Two of these layers were the VV and VH polarization bands from Sentinel-1, owing to the capabilities of SAR data for mapping palm plantations (Descals et al., 2019). The optical band 4 from Sentinel-2 (red band; wavelength centred at 665 nm) was the third input layer in the global oil palm layer. Band 4 was chosen because it is the 10-meter resolution band that provides the clearest depiction of harvesting trails in industrial plantations. In the red spectrum, harvesting trails have a high reflectance that contrasts with the low reflectance of the surrounding oil palm.

In contrast to industrial oil palm plantations, coconut palm plantations do not present a harvesting road network that can be identified in 10-meter satellite data. Extensive coconut palm plantations, such as those found in Tabou (Côte d'Ivoire) and in small islands such as Talina (Solomon Islands) or Mapun (Philippines), might present harvesting roads but these are not clearly
visible in Sentinel-1 and Sentinel-2. In addition, there were coconut palm plantations incorrectly classified as oil palm in the global oil palm layer (Descals et al., 2021), indicating that a spectral band other than band 4 could better distinguish oil palm from coconut palm. We also found in our preliminary analysis that sago forests (*Metroxylon sagu* Rottb.) and mango plantations (*Mangifera spp. L.*) could also be confused with coconut palm in the VV-VH-band 4 composites. Thus, we inspected the spectral separability between coconut palm, oil palm, sago palm, and mango plantations for all 10- and 20-meter
Sentinel-2 bands. To test the spectral separability, we collected 40 points for each tree species. We normalized the Sentinel-1 and -2 bands using the z-normalization and evaluated the separability using the one-dimensional Bhattacharyya distance (Theodoridis and Koutroumbas, 2006). The Bhattacharyya distance evaluates the overlap between two independent distributions; the higher the Bhattacharyya distance, the lower the overlap between the spectral values of coconut palm and another tree species.

The separability analysis revealed low separability between coconut palm and oil palm plantations in the VV and VH bands (Fig. A2), which indicates that Sentinel-1 may not be able to distinguish between oil palm and coconut palm. Among the spectral bands, Sentinel-2 band 11 (short-wave infrared spectrum; wavelength centred at 1,614 nm) exhibited the greatest spectral separability between coconut palm and oil palm in terms of Bhattacharyya distance. Since oil palm plantations
potentially overlap with coconut palm to a greater degree than mango and sago palm, we selected band 11 as the optical band for the classification of coconut palm. Since band 11 has a spatial resolution of 20 meters, we aggregated the Sentinel-1 composites to 20-meters using bilinear interpolation. As a result, the final coconut palm layer has a spatial resolution of 20 meters.

## 2.5 Semantic segmentation

The Sentinel-1 and Sentinel-2 composites were classified using a semantic segmentation model, specifically a U-Net model with MobileNetv2 as the backbone (Falk et al., 2019). Semantic segmentation is a type of deep learning model that consists of a pixel-wise classification of an image using a convolutional neural network. Semantic segmentation is well suited for mapping plantations, such as coconut palm, since it can automatically capture the spatial and contextual information in the image and, as a result, less effort is required compared to feature engineering in standard machine learning (Ma et al., 2019). Such
contextual information includes the shape of the plantation or texture patterns within the plantation.

Semantic segmentation models require image data with a fixed size for both training and prediction. We set the size of the input images to $512 \times 512$ pixels, which is approximately $10 \times 10$ km in a 20-meter resolution image. The collection of training data consisted of digitizing polygons in regions that were identified in the bioclimatic analysis. The polygons were drawn in

146 training images (Fig. 1b) using sub-meter resolution to discriminate coconut palm plantations from other land covers. The sub-meter resolution images were the images displayed as the base layer in Google Earth. The U-Net was used for binary classification of coconut palm (digitized polygons) and the rest of land covers (image background; see Fig. A3) and, thus, the resulting layer was a binary raster, in which each pixel presented values of 0 (coconut palms are not present) and 1 (coconut palms are present). In addition, we generated a probability layer using the second-last layers of the convolutional neural network. Rather than probability layers, the second-last layers represent a confidence score (ranging from 0 to 100) for each class prediction. The probability layer we provide corresponds to the second-last layer of the class 'coconut'. The U-net model was trained and deployed using the PyTorch framework in the Microsoft Planetary Computer hub.

## 2.6 Validation

We evaluated the accuracy of the global coconut palm layer using the good practices for estimating area and assessing accuracy as described by Olofsson et al. ( 2014). To assess the validity of the classification layer, we needed extensive, randomly distributed, well-characterised reference points across the coconut-producing region. We used a stratified random sampling over the areas delimited by the potential coconut palm distribution. A total of 10,200 reference points were sampled: 557 points in pixels classified as class 'coconut' and 9,643 points in pixels classified as class 'other'. In stratified random sampling, the pixels that present the same class have an equal probability of being sampled. Here, we sought a cost-effective alternative by visually reviewing the sub-meter resolution images from Google Earth because coconut palms can be identified using such data. The interpreters assigned a 'truth' label out of the following five interpretations:

0. Land cover could not be determined because sub-meter resolution data was not available.

1. Other land cover. Coconut palms are not present within the 20-meter pixel.

2. Sparse coconut palm. Low density of coconut palms. There are between 1 and 4 coconut palms within the 20-meter pixel.

3. Dense open-canopy coconut palm. There are more than 4 coconut palms within the 20-meter pixel, but coconut palms do not reach the full canopy closure.

4. Closed-canopy coconut palm. There are more than 4 coconut palms within the 20-meter pixel and coconut palms fully cover the ground.

The validation points were first labelled by a team of three interpreters, and then we used a second level of verification (Szantoi et al., 2021). The second level of verification consisted of an independent interpreter that verified the points that the team labelled 'coconut'. There were 1,814 points in which the land cover could not be determined and, thus, the total number of reference points was 10,186 in the accuracy assessment (Fig. 1c). The number of points was 7,581 for 'other land cover', 164 for 'sparse coconut', 120 for 'dense open-canopy coconut', and 202 for 'closed-canopy coconut'. In the accuracy assessment, the points labelled 'other land cover' were recoded as 0 (class 'other'). For the class 'coconut', we considered three definitions (Fig. 2a). The first definition assigned class 'coconut' when, at least, one coconut palm was found within a 20-meter pixel.

Points labelled as 'sparse coconut', 'dense open-canopy coconut', and 'closed-canopy coconut' were recoded as 1. This initial definition aimed to provide an estimate of all coconut-producing regions. The second definition considered as class 'coconut' the points labelled as 'dense open-canopy coconut' and 'closed-canopy coconut'. This second definition aimed to evaluate the capability of Sentinel-1 and -2 for mapping dense coconut stands that do not reach full canopy closure. The third definition only considered points labelled 'closed-canopy coconut' in the class 'coconut'.

The accuracy metrics included the producer's accuracy (PA), the user's accuracy (UA), and the overall accuracy (OA). The producer's accuracy represents the proportion of pixels of a given class that were not omitted in the classification, while the user's accuracy shows the proportion of pixels that were not committed for a given class. The OA represents the proportion of pixels that were correctly classified. We also tested whether the OA was significantly higher than the no-information rate. The no-information rate is the overall accuracy obtained by classifying all pixels with the largest land cover class —in our case, the class 'Other'. An overall accuracy significantly higher than the no-information rate indicates that the classification model did better than classifying indiscriminately all pixels with the class 'Other'. We reported the post-stratified metrics for PA, UA, and OA using the practices in Olofsson et al. (2014) and Szantoi et al. (2021). These practices also explain the area estimation for each class in the land cover map. While the mapped area reveals the area that was classified as a particular class, the area estimates account for omission and commission errors and provide an area with a confidence interval. All metrics of accuracy and area estimates were reported with a confidence interval of 95 %.

## 2.7 Area estimates for small tropical islands

The global coconut palm layer relies on the availability of Sentinel-1 and Sentinel-2 data. These two satellites provide images for the larger land masses across the globe, but the data is missing in parts of the Pacific and other small tropical islands. On small islands with no Sentinel-1 or Sentinel-2 data, coconut palm mapping was not possible using our classification model. To overcome this issue, we used a sampling-based method to estimate the coconut palm area in small tropical islands, owing to the availability of sub-meter resolution satellite images in most of these islands. The sampling-based approach comprised randomly sampling 5,000 points within the small tropical island extents (Fig. 1d). Small tropical islands included islands with an area between 1 and 200 ha in the tropics (latitudes within latitudes 30°S and 30°N) in a reference dataset (Sayre et al., 2019). The points were visually interpreted and categorized into the following five classes:

0. Land cover could not be determined because sub-meter resolution data was not available.

1. Non-vegetated land cover. Vegetation coverage is <50 % and coconut palms are not present within a 20-meter bounding box.

2. Other vegetation. Vegetation coverage is >50 % and coconut palms are not present within a 20-meter bounding box.

4. Sparse coconut palm. Low density of coconut palm; between 1 and 4 coconut palms within a 20-meter bounding box.

5. Dense open-canopy and closed-canopy coconut palm; more than 4 coconut palms within a 20-meter bounding box.

The area occupied by coconut palm in the small islands was inferred using the proportion of *'coconut'* points ($n_{coconut}/n_{total}$); $Area_{coconut} = Area_{islands} \times n_{coconut}/n_{total}$, where $Area_{coconut}$ is the area covered by coconut palm and $Area_{islands}$ is the total area of small islands per country or globally. The 95 % confidence interval for $Area_{coconut}$ was estimated using the confidence interval

for a population proportion; $CI = 1.96 \times \sqrt{p(p-1)/n}$, where $CI$ is the confidence interval, $p$ is the proportion of points categorized as 'coconut' ($n_{coconut}/n_{total}$), and $n$ is the total number of sampled points. The area estimates for small islands did not consider the difference between dense open-canopy and closed-canopy coconut palm. The distinction was made solely to assess the performance of the classification model for mapping dense open-canopy coconut palm.

## 3 Results

We collected 1,139 points in places where coconut palms were visually identified using sub-meter resolution images (Fig. 1a). The points were located in the tropics between 25.24°S and 26.40°N, generally in low-elevation areas close to the coast. The coconut palms at the highest elevation were found at 988 meters in the Indian state of Karnataka. Nevertheless, the average altitude was 101 meters and the average distance to the ocean was 750 meters. Some coconut palms were found hundreds of kilometres inland; for example, a coconut palm was found in Bolivia at 808 kilometres from the Pacific Ocean (Fig. A4a).

These coconut palms presented yellow-coloured leaves, indicating substandard growing conditions, and were never observed as a plantation. The bioclimatic analysis confirmed that coconut palm grows predominantly in regions with a warm and humid oceanic climate, characterized by low daily and seasonal temperature variations due to the proximity of oceans. In the 1,139 points used for the bioclimatic analysis, the mean annual temperature ranged from 22.4 °C (minimum) to 28.8 °C (maximum) (Table A1). The lowest monthly mean temperature recorded during the coldest month was 11.5 °C. The southern and northern

limits of the potential coconut palm distribution were predominantly semi-arid regions (Fig. A5). We found that coconut palm is cultivated in a variety of rainfall regimes. Coconut palm plantations were found in arid and semi-arid regions (annual rainfall <250 mm), such as Dhofar Governorate in Oman (17.0054°N, 54.1069°E), Sindh Province in Pakistan (24.7204°N, 67.5855°E), and Tumbes Province in Peru (4.0481°S, 80.9472°W). However, coconut palm is grown with irrigation in these regions and represents a negligible area compared to the extensive plantations in Kerala State in India, the Philippines, and

Indonesia, where rainfall is abundant (annual rainfall >2000 mm).

The global coconut palm layer has an overall accuracy of 99.04 ± 0.21 % (intervals represent 95 % confidence), based on the post-stratified accuracy assessment of the 10,186 validation points and considering the first definition of coconut palm, which included sparse coconut palm, and dense open- and closed-canopy coconut palm. The overall accuracy was greater than the

no-information rate (94.13 ± 0.51%), indicating that the classification improved upon one in which all pixels were classified as class 'other'. The producer's and user's accuracy were 11.30 ± 2.33 % and 79.21 ± 3.46 % for the class 'coconut',

respectively, and 99.97 ± 0.01 % and 99.07 ± 0.21 % for the class 'other' (Table 1). Without considering points in sparse coconut palm, the producer's accuracy increased to 32.32 ± 10.17 %. If only closed-canopy coconut palm were considered, the producer's accuracy was 71.51 ± 23.11 % for the class 'coconut'. This large difference in producer's accuracy for the
different definitions of class 'coconut' indicates that sparse and dense open-canopy coconut palm were largely omitted in the classification.

According to a visual examination of sub-meter satellite images, we identified several palm tree species that were incorrectly classified as class 'coconut' (Fig. 2b), explaining the low user's accuracy for the class 'coconut' (79.2 %). We found false
positives in sago palm and nypa palm (*Nypa fruticans* Wurmb.) in Southeast Asia and the Pacific, raffia palm (*Raphia spp.* P.Beauv.) in South America and Africa, areca palm (*Areca catechu* L.) in India, euterpe palm (*Euterpe edulis* Mart.) in South America, attalea palm (*Attalea spp.* Kunth) in Central America, and palmyra palm (*Borassus spp.* L.) in Africa. Even though band 11 was included in the classification model, oil palm plantations, especially those of smallholders, were residually detected as coconut palm. Most of these false positives were eliminated in the final layer by manually editing the output of the
classification (Fig. 3 and Fig. A6). The vast bulk of these palms were found apart from coconut palm plantations and could be identified in the high-resolution satellite data. For instance, in New Guinea, coconut palm typically covers the first kilometres from the sea, while sago palm covers areas farther inland (Fig. A7). We also found false positives for class 'coconut' in mango, a non-palm plantation. Mango plantations were located on the Pacific coast of Mexico and in Gujarat State, India. Removing false positives in mango plantations was problematic due to the co-occurrence of coconut palm and mango plantations in the
landscape. In addition, we found plantations that contained both mango and coconut palm (Fig. A4b). Other intercropping settings were found with maize (*Zea mays* L.), rice (*Oryza spp.* L.), and banana (*Musa spp.* L.) (Fig. A4c, d, and e). In contrast, we did not find intercropping in closed-canopy coconut palm, which was generally devoid of understory aside from grasslands and small shrublands (Fig. A4f and g).

Most of the validation points for the class 'coconut' fell in the three main coconut-producing regions: the Philippines (115 points), Indonesia (130 points), and India (162 points). Owing to this dense sampling, we could generate separate accuracy assessments for these three countries (Table 1). The next country was Sri Lanka, with only 14 points labelled as class 'coconut', which is insufficient for evaluating the accuracy. The accuracy assessment revealed similar omission rates for closed-canopy coconut palm at the country level compared to the global assessment. The producer's accuracy for the class 'coconut' was
lowest in the Philippines (70.25 ± 29.18 %), compared to Indonesia (80.03 ± 31.37 %) and India (77.23 ± 34.56 %), although the large confidence interval indicates that the difference is not significant.

The total area mapped as coconut palm was 5.55 x $10^6$ ha (Table 2). Coconut palm was mainly found in India and Southeast Asia (Fig. 4), regions where we also found most of the largest clusters of coconut palm plantations (Fig. 5 and Fig. A8, which
depicts the coconut palm probability layer). The area estimates revealed that coconut palm covers 38.93 ± 7.89 x $10^6$ ha,

including sparse, dense open-, and closed-canopy coconut palm. If only dense open- and closed-canopy coconut palm was considered in the accuracy assessment, the global coconut palm area is $12.66 \pm 3.96$ x $10^6$ ha, which is similar to the 11.61 x $10^6$ ha reported globally by FAO. The coconut palm mapped area was 1.54 x $10^6$ ha in the Philippines, 1.73 x $10^6$ ha in Indonesia, and 1.29 x $10^6$ ha in India, which together represent 82 % of the global coconut palm mapped area (Fig. A9). Other hotspots of coconut production were found along the Pacific coast of Mexico, Brazil, Ghana, Côte d'Ivoire, Tanzania, Mozambique, Sri Lanka, Vietnam, Thailand, and Papua New Guinea. In some of these countries, the mapped coconut palm area corresponded well with FAO statistics, for instance, in Papua New Guinea, Vietnam, and Thailand. In contrast, Tanzania is the fourth largest coconut-producing country with 0.60 x $10^6$ ha according to FAO, but only 0.03 x $10^6$ ha were mapped. In East Africa, coconut palm is sparsely planted (Fig. A4h), which could account for our likely underestimation. The coconut palm area estimate for Tanzania ($0.52 \pm 0.48$ x $10^6$ ha) was consistent with FAO, although the estimate has a large confidence interval due to low sampling in this country.

We found that several countries in the Pacific Ocean had a large coconut palm area in comparison to their overall land area. Papua New Guinea had the largest coconut palm area mapped (0.17 x $10^6$ ha), followed by Vanuatu (0.6 x $10^6$ ha) and the Solomon Islands (0.5 x $10^6$ ha). Fig. 4 shows the availability of Sentinel-1 and Sentinel-2 data, which is lacking in many islands in the Pacific Ocean. According to sampling-based estimates in small tropical islands (land area < 200 ha), Indonesia and the Philippines were the countries with the largest coconut palm area (Fig. A10a), accounting for $33,798 \pm 530$ ha and $21,231 \pm 630$ ha of dense coconut palm and $34,944 \pm 556$ ha and $16,681 \pm 444$ ha of sparse coconut palm, respectively. The ratio of coconut palm to total area revealed that small islands in the Pacific had the highest coconut palm area relative to land area (Fig. A10b). Tuvalu had the highest percentage, with 81 % of the land in small islands covered with coconut palm. Other small islands in the Pacific countries presented a low proportion of coconut palm relative to total area, but a high proportion relative to vegetated land. In French Polynesia, the overall proportion of coconut palm was only 22 %, but it comprised 50 % of all vegetated areas in the small islands.

## 4 Discussion

We produced the first global coconut palm layer with a 20-meter resolution and estimated the global area of coconut palm using remotely sensed data for the year 2020. We also generated a probability layer that provides a score indicating the confidence level of the model output. This probability layer could serve as a proxy for coconut palm density. The global coconut palm layer demonstrates the capabilities of Sentinel-1 and Sentinel-2 to map coconut palm. We observed that the spectral separability in band 11 was imperfect as residual false positives were still occurring in oil palm, sago palm, and other palm species, explaining the low user's accuracy for the class 'coconut' (79.2 %). Our model omitted most of the coconut palm that did not reach full canopy closure, and coconut palm remained broadly undetected when trees were sparsely distributed throughout the land. This issue was also found in industrial plantations with a wide planting mark. A similar

problem was found in the global mapping of oil palm, which reported higher omission errors in semi-wild oil palm in West Africa (Descals et al., 2021). Despite this, the producer's accuracy for closed-canopy coconut palm ($71.51 \pm 23.11$ %) was

similar to those obtained in the global oil palm layer, which were $75.78 \pm 3.55$ % for smallholders and $86.92 \pm 5.12$ % for industrial oil palm. This indicates that Sentinel-1 and Sentinel-2 can map closed-canopy palm species with a similar accuracy.

Sub-meter resolution images could be used in future research to accurately map sparse coconut palm in small tropical islands where Sentinel-1 and -2 data are unavailable. Object detection using deep learning applied to very high-resolution images (<1

meter), such as those obtained by DigitalGlobe or Planet, offers great potential for the detection of individual coconut palms (De Souza and Falcão, 2020; Vermote et al., 2020; Freudenberg et al., 2019). This approach could be used to detect coconut palm plantations with incomplete canopy closure and coconut palms that are scattered across the land. In our study, the decametric resolution of Sentinel-1 and Sentinel-2 images made the use of object detection techniques unfeasible. Object detection using deep learning and sub-meter images could complement our closed-canopy coconut palm layer and could also

be useful for mapping different palm trees, including coconut palm, oil palm, and sago palm. Because of the high costs of such imagery, sub-meter resolution mapping would only be feasible in specific areas where these high-resolution data are crucial for informing planning and decision-making about land use and agricultural development.

The potential coconut palm distribution confirmed previous insights about coconut growing requirements, with an area

covering most tropical coastal regions but not those with high aridity or low temperatures. Our potential distribution coincides with the coastal areas in a similar map for coconut palm (Coppens D'Eeckenbrugge et al., 2018). Soil types were not considered in the bioclimatic analysis for the estimation of the potential coconut palm distribution. Coconut palm prefers sandy soils, but different types of soil can support the growth of coconut palm as long as they are well-drained (Chan and Elevitch, 2006), which explains why coconut palm grows in the first few kilometres of coastline in Papua, while sago palm dominates the

landscape in inland swampy areas. The drainage requirements for coconut cultivation also explain the unsuitability of vertisols, also known as black soils, which contain a high content of expansive clay minerals with inherent poor drainage. Despite not including a soil map in the bioclimatic analysis, the resulting layer from the coconut palm classification presented a negligible overlap with vertisol areas, for instance the Deccan Traps in India. Additionally, we found that coconut palm generally grows in low-elevation coastal regions, but also found coconut palm in mountainous regions in Tanzania, India and, especially, the

Philippines, corroborating previous observations in the country (Pabuayon et al., 2008). We did not include areas more than 200 kilometres from the coast because we found very few coconut palms beyond that 200-kilometer threshold in our visual assessment of high-resolution images.

Our findings show that the area designated for growing sparse, open-canopy, and closed-canopy coconut palm ($38.93 \pm 7.89$

x $10^6$ ha) is significantly larger than the area recognized by the FAO ($11.6$ x $10^6$ ha). The FAO underreports planted area because it is based on production data and yield, and it does not account for areas sparsely covered in coconut palm. This

finding indicates that much more land has been allocated to coconut palm growing than previously reported, even though coconut production may not be very important on much of that land. We do not know enough about the nature of sparsely planted coconut areas to judge how productive these lands are. In areas where coconut palm is intercropped with other crops,
overall land productivity depends on more than coconut production. Sparse coconut palm areas may also relate to old plantations with limited maintenance and low productivity, which is a known problem in the coconut industry (Peiris et al., 2001). Overall, the coconut industry is known to have a gap between potential and actual yields, which relates to the prevalence of pests and diseases, inferior varieties, out-dated agronomical practices, and the high proportion of senile palms (Alouw and Wulandari, 2020). Therefore, the large area of sparse and dense open-canopy coconut palm indicates that production increases
can likely be achieved on the existing lands allocated to coconut production.

The potential increases in coconut production have environmental consequences because demand for coconut products is rapidly growing, putting pressure on the industry to expand land holdings. Global coconut revenues are predicted to increase from US$ 5.7 billion in 2022 to USD 7.4 billion in 2027 (MarketsandMarkets, 2022), and the more production increases that
can be met on existing land, the less impact this will have on food security and biodiversity in areas that would otherwise be displaced by new coconut palm plantations. Furthermore, our map will help in predicting the likely impact of climate change on coconut productivity, as recently determined for India (Hebbar et al., 2022). While we acknowledge that the impacts of these production predictions remain unclear, having the first high-resolution map of global coconut palm provides a solid basis for monitoring how this crop develops. This map also allows for the quantification of the effects of coconut palm expansion
on natural ecosystems such as tropical lowland forests, mangroves, and beach forests, which helps to inform global biodiversity and environmental policy. Such policies could focus on increasing productivity on existing coconut lands so that no new expansion is required, potentially focusing on sparse and open coconut palm regions where yield increases might be less challenging. On the other hand, meeting coconut production increases on existing dense coconut land could also allow for phasing out unproductive sparse coconut lands and restoring them to natural ecosystems with potential biodiversity and other
environmental benefits (Carr et al., 2021).

While we were unable to map coconut palm in small islands in the Pacific (because of the absence of Sentinel-1 and -2 data), our area estimates confirm that coconut palm is a dominant species in many of these island nations, with several countries having more than half of their land area of small islands covered in coconut palm. This indicates the importance of this crop
for many smallholder producers in the Pacific, who often grow this cash crop together with other crops, with coconut palm being the permanent crop and other crops grown when their prices are high (Feintrenie et al., 2010). Like elsewhere, these smallholder producers struggle with low coconut productivity, but this may be compensated by good yields from other crops. Where coconut palm is grown as a monoculture, reorganization of the coconut industry has been proposed, potentially along similar lines as palm oil production, based on the model Nucleus Estate/Nucleus-Plasma concept. High coconut palm coverage
on small islands in the Pacific and Indian Ocean and to a lesser extent in the Caribbean, may be a significant threat to

biodiversity and other ecosystem services (Meijaard et al., 2020), especially because coconut palm can be invasive in tropical islands (Young et al., 2017). More work needs to be done to map coconut palm areas on these islands, ideally using sub-meter resolution data where Sentinel-1 and -2 data are currently unavailable. Once such maps become available, they can provide better insight into the extent to which coconut palm has displaced natural ecosystems, relative coconut productivity (in areas

with detailed harvest information), and the potential for coconut palm expansion, conversion to other forms of agriculture, or restoration of natural ecosystems. Detailed and accurate spatial information is a key component in any land-use optimization planning, for coconut palm as well as other crops.

## 5 Data availability

The dataset presented in this study is freely available for download at https://doi.org/10.5281/zenodo.8128183 (Descals, 2022).

The file 'GlobalCoconutLayer_2020_v1-2.zip' contains 878 raster tiles of 100x100 km in geotiff format. The raster files are the result of a convolutional neural network that classified Sentinel-1 and Sentinel-2 annual composites into a coconut palm layer for the year 2020. The images have a spatial resolution of 20 meters and contain two classes:

        [0] Other land covers that are not coconut palm.

        [1] Coconut palm.


The file 'GlobalCoconutLayer_2020_densityMap_1km_v1-2.zip' contains the 20-meter coconut palmlayer aggregated to 1 km. The value of each pixel represents the coconut palm area (in squared meters) within the 1-km pixel.

The file 'Validation_points_GlobalCoconutLayer_2020_v1-2.shp' includes the 10,200 points that were used to validate the

product. Each point includes the attribute 'Class', which is the class assigned by visual interpretation of sub-meter resolution images, and the attribute 'predClass', which reflects the predicted class by the convolutional neural network. The 'predClass' values are the same as the raster files:

        [0] Other land covers that are not coconut palm.

        [1] Coconut palm.

The attribute 'Class' contains the following values:

        [0] Land cover could not be determined because sub-meter resolution data was not available.

        [1] Other land covers that are not coconut palm.

        [2] Sparse coconut palm. Low density of coconut palms; between 1 and 4 coconut palms within the 20-meter pixel.

        [3] Dense open-canopy coconut palm; more than 4 coconut palms within the 20-meter pixel but coconut palms do not

reach the full canopy closure.

        [4] Closed -canopy coconut palm; more than 4 coconut palms within the 20-meter pixel and coconut palms fully cover the ground.

[5] Palm species that are not coconut palm.

The global coconut palm layer, the probability layer for class 'coconut', and the coconut palm density map can be visualized online at: https://adriadescals.users.earthengine.app/view/global-coconut-layer (last access: 6 July 2022).

The Sentinel-1 SAR GRD and Sentinel-2 Level-2A used in this study are available at the Copernicus Open Access Hub: https://scihub.copernicus.eu/ (last access: 6 July 2022). We used all Sentinel-1 and Sentinel-2 images that overlapped the 430 potential distribution of coconut palm for the year 2020.

The WorldClim bioclimatic variables (WorldClim V1 Bioclim) (Hijmans et al., 2005) can be accessed at https://www.worldclim.org/data/v1.4/worldclim14.html (last access: 6 July 2022).

Very high-resolution images (spatial resolution <1 m) from DigitalGlobe can be visualized in the Google Earth Engine code editor or Google Maps.

The 5 arcmin global coconut palm area modelled with SPAM (Yu et al., 2020) is available at https://doi.org/10.7910/DVN/PRFF8V.

The country-wide harvested area of coconut palm was extracted from the FAOSTAT database at http://www.fao.org/faostat/en/ (accessed on 10 March 2022) under the item 'Coconuts in shell - Crops and livestock products (Production)' (FAO, 2022).

## 6 Code availability

The original code of the U-Net model can be found at: https://github.com/qubvel/segmentation_models.pytorch (Iakubovskii, 445 2019)

## 7 Conclusions

We mapped the global distribution of coconut palm using a deep learning model that classified satellite data (SAR Sentinel-1 and optical Sentinel-2) into a 20-meter land cover map depicting the extent of closed-canopy coconut palm. The model achieved a high accuracy for closed-canopy coconut palm, and the resulting coconut palm layer accurately depicts the regions 450 with the highest density of coconut palm. The presented dataset can be integrated into the recently published Essential Agricultural Variables' "Perennial Cropland Mask" as well as the Food and Agricultural Organization's LCCS classification as "Cultivated and Managed Terrestrial Areas" - "Tree Crops" (Di Gregorio, 2005).

Our global coconut palm layer study provides the accurate high-resolution data required to evaluate the relationships between
vegetable oil production and the synergies and trade-offs between different sustainable development goal indicators. Moreover, the global coconut palm layer can be used in geospatial analysis to assess the spatial overlap between coconut palm extent and areas of highly threatened species, species endemism, and species richness. In this regard, the coconut palm map presented in this study can be valuable for studying the environmental impacts associated with coconut cultivation in biodiversity hotspots.

## 8 Author contributions

The conceptualization for this work originated from SW, ZS, MS, and EM. AD designed the study. AD, RD, TA, and NU collected the training data and RD, ZH, TA, NU collected the reference points. AD implemented the data processing workflow and generated the figures and tables. AD, SW, ZS, and EM wrote the draft and AD, SW, ZS, MS, RD, ZH, TA, NU, DLAG, and EM were involved in the revision of the manuscript.

## 9 Competing interests

The authors have no conflicts of interest to declare.

## 10 Financial support

We acknowledge funding from the Microsoft AI for Earth program.

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

Table 1: Accuracy assessment of the global coconut palm layer for the year 2020. The accuracy metrics were estimated with 10,186 points randomly distributed in the regions where coconut palm can potentially grow. The accuracy metrics are reported with a 95 % confidence interval.

| | Overall accuracy[1] (%) | User's accuracy Coconut[1] / Other (%) | Producer's accuracy Coconut[1] / Other (%) | Producer's accuracy Coconut[2] (%) | Producer's accuracy Coconut[3] (%) |
|---|---|---|---|---|---|
| World | 99.04 ± 0.21 | 79.21 ± 3.46 / 99.07 ± 0.21 | 11.30 ± 2.33 / 99.97 ± 0.01 | 32.32 ± 10.17 | 71.51 ± 23.11 |
| Philippines | 93.85 ± 2.82 | 84.75 ± 6.52 / 94.12 ± 2.89 | 29.66 ± 10.39 / 99.53 ± 0.20 | 53.06 ± 21.83 | 70.25 ± 29.18 |
| Indonesia | 98.99 ± 0.61 | 85.82 ± 5.78 / 99.06 ± 0.61 | 33.21 ± 14.51 / 99.92 ± 0.03 | 58.75 ± 27.47 | 80.03 ± 31.37 |
| India | 95.90 ± 1.61 | 74.73 ± 6.33 / 95.24 ± 1.79 | 12.88 ± 4.32 / 99.75 ± 0.06 | 36.87 ± 18.69 | 77.23 ± 34.56 |

[1]Sparse, dense open-, and closed-canopy coconut palm

[2]Dense open- and closed-canopy coconut palm

[3]Closed-canopy coconut palm

Table 2: Coconut palm area mapped for 2020, harvested area obtained from FAO statistics for 2020, and area estimates for three definitions of coconut palm: 1) sparse, dense open-, and closed-canopy coconut palm, 2) dense open- and closed-canopy coconut palm, and 3) only closed-canopy coconut palm. The area estimates are reported with a 95 % confidence interval.

| | Coconut area mapped (ha x $10^6$) | Coconut area FAO 2020 (ha x $10^6$) | Coconut area estimate[1] (ha x $10^6$) | Coconut area estimate[2] (ha x $10^6$) | Coconut area estimate[3] (ha x $10^6$) |
|---|---|---|---|---|---|
| World | 5.55 | 11.61 | 38.93 ± 7.89 | 12.66 ± 3.96 | 5.03 ± 1.65 |
| Philippines | 1.54 | 3.65 | 4.41 ± 1.53 | 2.29 ± 0.95 | 1.46 ± 0.63 |
| Indonesia | 1.73 | 2.77 | 4.38 ± 1.91 | 2.38 ± 1.12 | 1.64 ± 0.66 |
| India | 1.29 | 2.15 | 7.46 ± 2.44 | 2.46 ± 1.24 | 0.79 ± 0.11 |

[1]Sparse, dense open-, and closed-canopy coconut palm

[2]Dense open- and closed-canopy coconut palm

 [3]Closed-canopy coconut palm

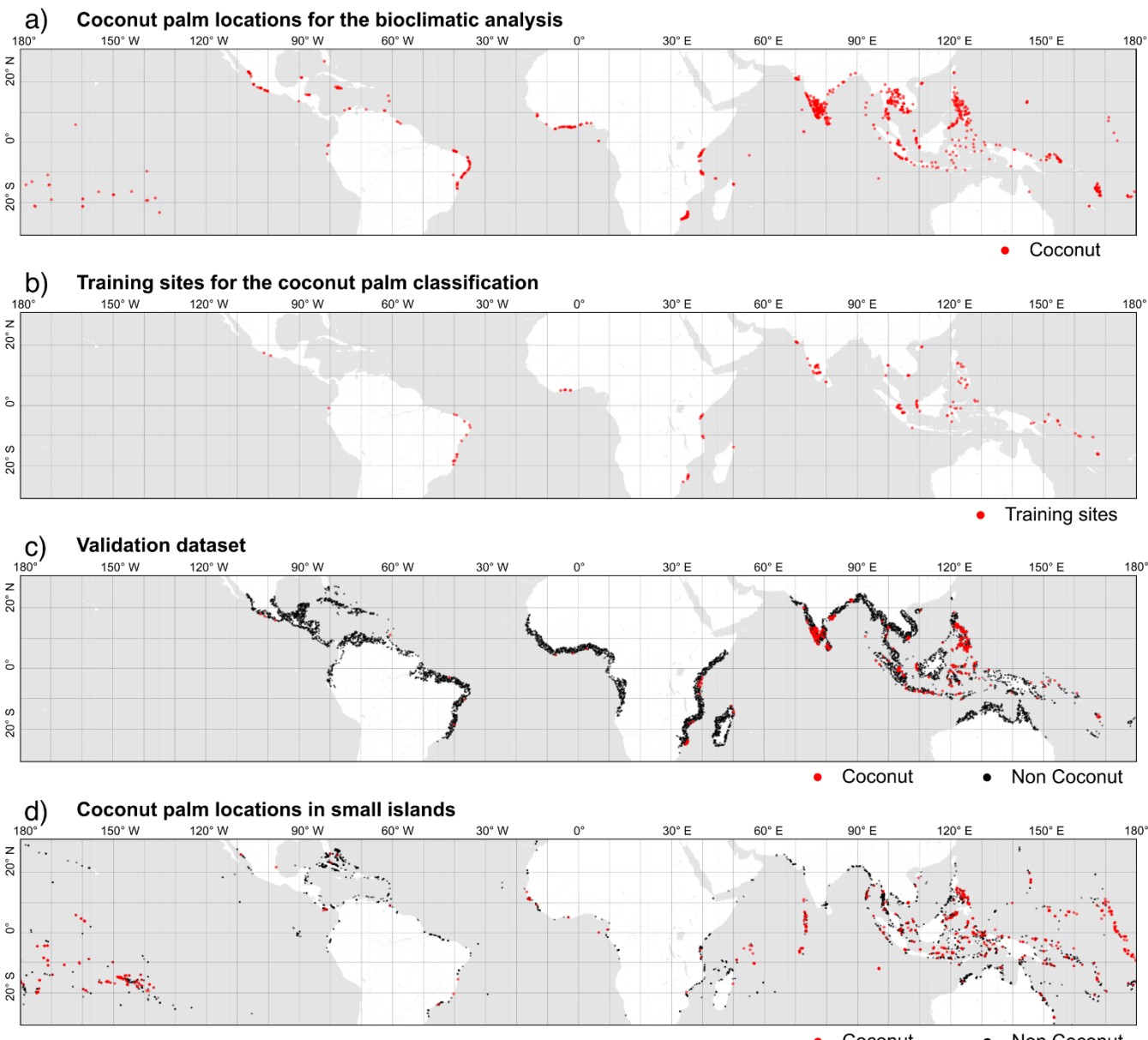


**Figure 1: Four point datasets used in the methodology. (a) 1,139 points depicting coconut palm locations found by visual inspection of sub-meter satellite images. These points were used in a bioclimatic analysis to determine the potential distribution area of coconut palm. (b) Location of the 146 training sites. In these locations, Sentinel-1 and Sentinel-2 annual composites were labelled in 10x10 km for training a semantic segmentation model. (c) Validation dataset generated from a stratified random sampling. The dataset**

**consists of 10,186 points and was used to evaluate the accuracy of the global coconut palm layer and estimate the global coconut palm area. (d) 5,000 points randomly sampled in small tropical islands (areas from 1 to 200 ha). The points were used to estimate the coconut palm area in small islands, where Sentinel-1 and Sentinel-2 might not be available.**

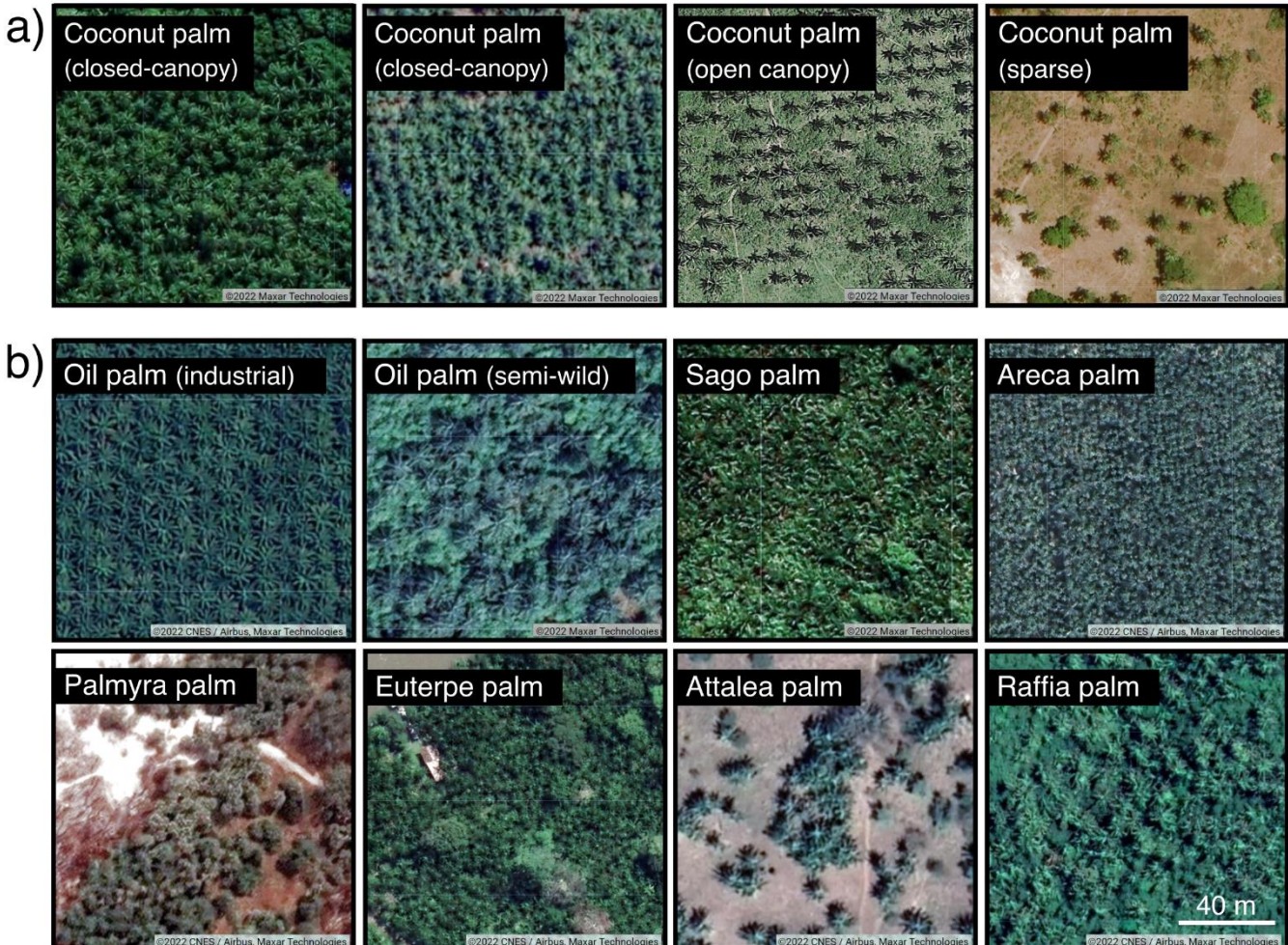


**Figure 2: Sub-meter resolution images depicting (a) coconut palm and (b) other palm species found in the tropics. The images show (from left to right and form up to down) a closed-canopy coconut palm stand in Papua New Guinea (6.124043°S, 134.13848°E) and Indonesia (1.077958°N, 108.966256°E), dense open-canopy coconut palm in Philippines (13.792082°N, 123.016486°E), sparse coconut palm in Kenya (4.367173°S, 39.493028°E), industrial oil palm in Indonesia (1.123642°N, 100.498538°E), semi-wild oil palm in Nigeria (6.641218°N, 5.388639°E), sago palm forest in Papua New Guinea (6.122091°S, 134.139178°E), areca palm in India (13.980709°N, 75.632272°E), palmyra palm in Gabon (6.078832°S, 12.330894°E), euterpe palm in Brazil (1.492261°S, 48.3734988°W), attalea palm in Mexico (16.10187°N, 97.396666°W), and raffia palm in Brazil (4.295997°S, 42.943344°W). The satellite images are the sub-meter resolution images that are displayed as the base layer in Google Earth @ Google.**


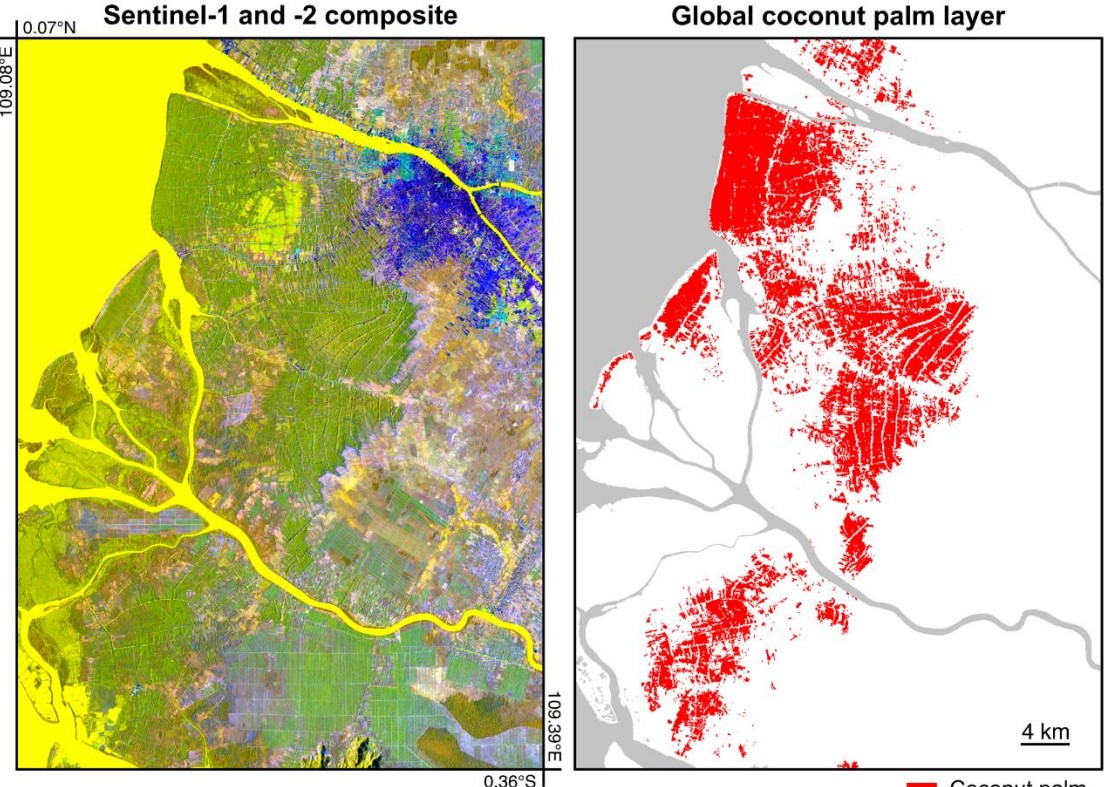


**Figure 3: Classification of a Sentinel-1 and Sentinel-2 annual composite into a land cover map of coconut palm in West Kalimantan (Indonesia). The Sentinel-1 and -2 composite (left panel) includes the polarization bands VV and VH, and the spectral band 11 (short-wave infrared). In this composite, coconut palm and oil palm appear in different shades of green. Oil pam is present in the lower-right part of the image with a brighter green colour than coconut palm. In this composite, water appears in yellow. The**
**classification image (right panel) shows the coconut palm plantations in red.**


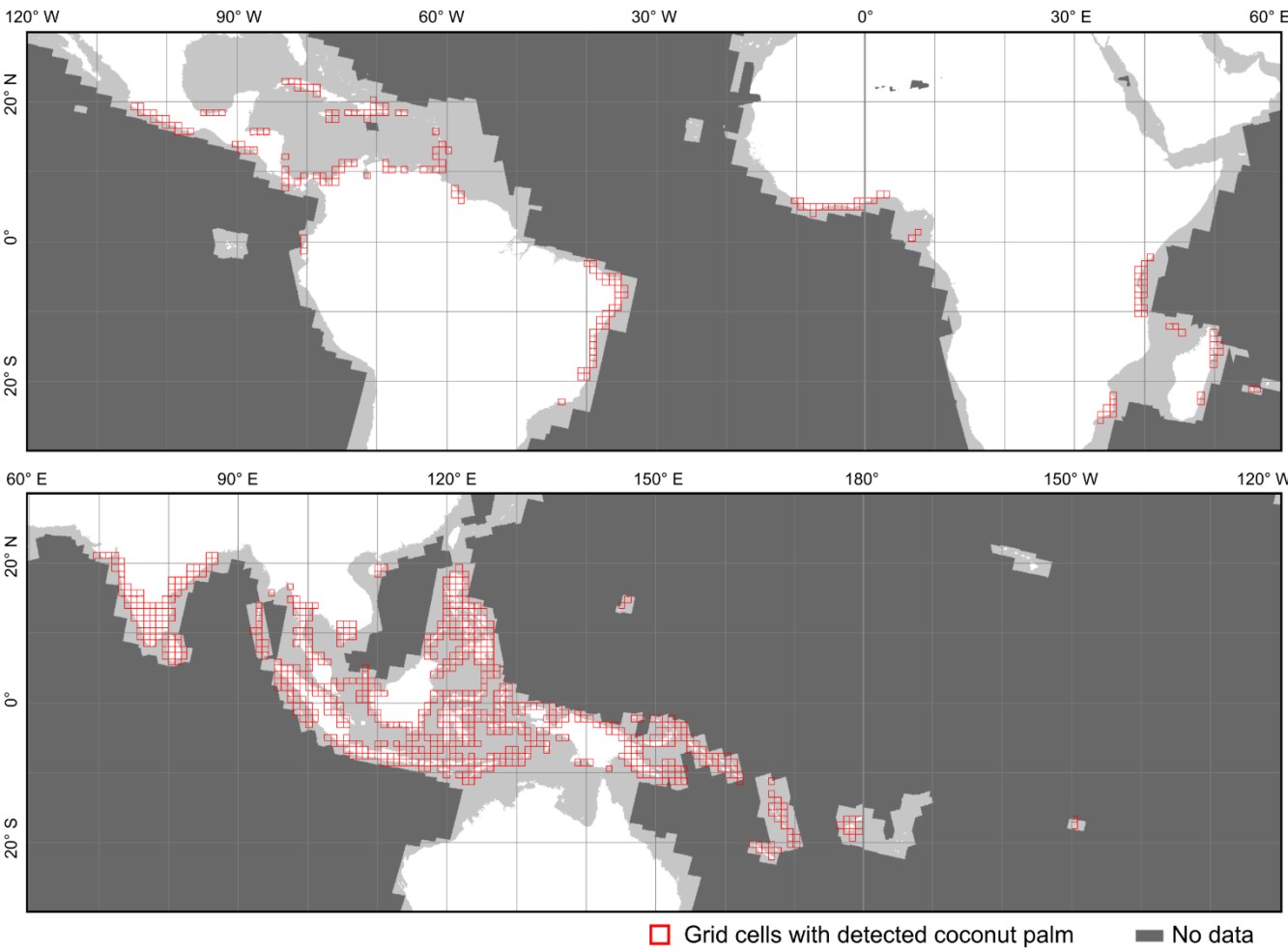

**Figure 4: Global occurrence map of coconut palm. Grid cells in red depict areas where coconut palm was detected using a U-Net model and annual Sentinel-1 and Sentinel-2 composites for 2020. The cell size is 100 x 100 km. Dark grey represents areas where Sentinel-1 or Sentinel-2 were not available.**

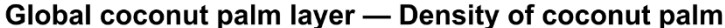

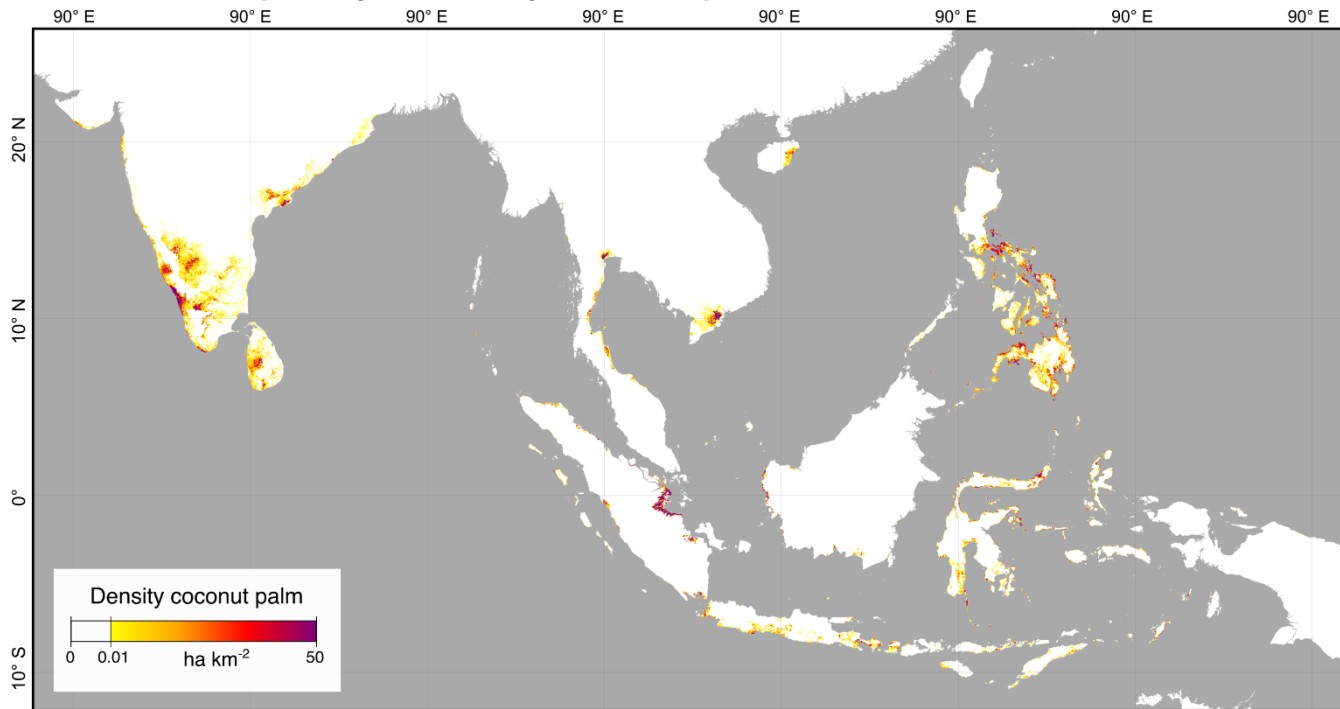

**Figure 5: Density of coconut palm in India and Southeast Asia at one-kilometer resolution. The map was generated using the 20-m global coconut palm layer. The density map highlights the primary regions of coconut production.**




# APPENDIX A

**Table A1: Range of climate values extracted from 1,139 coconut palm locations in the world. These ranges represent the minimum and the maximum values of the 19 WorldClim bioclimatic variables, elevation, slope, and maximum distance to the sea. The variable names bio05 and bio06 represent the maximum temperature of the warmest month and the minimum temperature of the coldest month. Variable names in bold present a low collinearity and were used in the bioclimatic analysis for estimating the potential coconut palm distribution.**

| Variable name | Minimum value | Maximum value | Unit |
|---|---|---|---|
| Annual mean temperature | 22.4 | 28.8 | °C |
| **Mean diurnal range** | 4.1 | 12.2 | °C |
| **Isothermality** | 39.0 | 93.0 | % |
| Temperature seasonality | 1.1 | 37.8 | °C |
| Max temperature of warmest month | 28.3 | 37.7 | °C |
| Min temperature of coldest month | 11.5 | 25.1 | °C |
| Temperature annual range (bio05-bio06) | 5.4 | 22.3 | °C |
| **Mean temperature of wettest quarter** | 21.6 | 29.2 | °C |
| **Mean temperature of driest quarter** | 19.1 | 29.7 | °C |
| Mean temperature of warmest quarter | 23.7 | 31.4 | °C |
| Mean temperature of coldest quarter | 18.2 | 27.7 | °C |
| Annual precipitation | 108 | 5132 | mm |
| **Precipitation of wettest month** | 35 | 1427 | mm |
| Precipitation of driest month | 0 | 282 | mm |
| **Precipitation seasonality** | 8 | 165 | Coef. of variation |
| Precipitation of wettest quarter | 83 | 3377 | mm |
| Precipitation of driest quarter | 0 | 935 | mm |
| **Precipitation of warmest quarter** | 28 | 1268 | mm |
| **Precipitation of coldest quarter** | 0 | 2776 | mm |
| Elevation | 0 | 988 | m |
| **Slope** | 0 | 26.4 | ° |
| Maximum distance to sea | 0 | 278 | km |



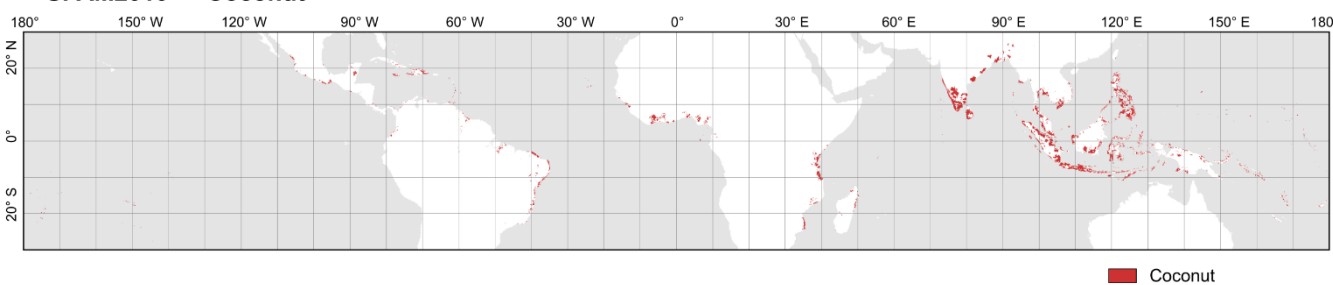

**Figure A1: Coconut palm map extracted from the Spatial Production Allocation Model for 2010 (SPAM2010). The layer represents areas where the extent of coconut palm plantations exceeded 50 hectares within each 5-arcmin grid of the SMAP dataset.**


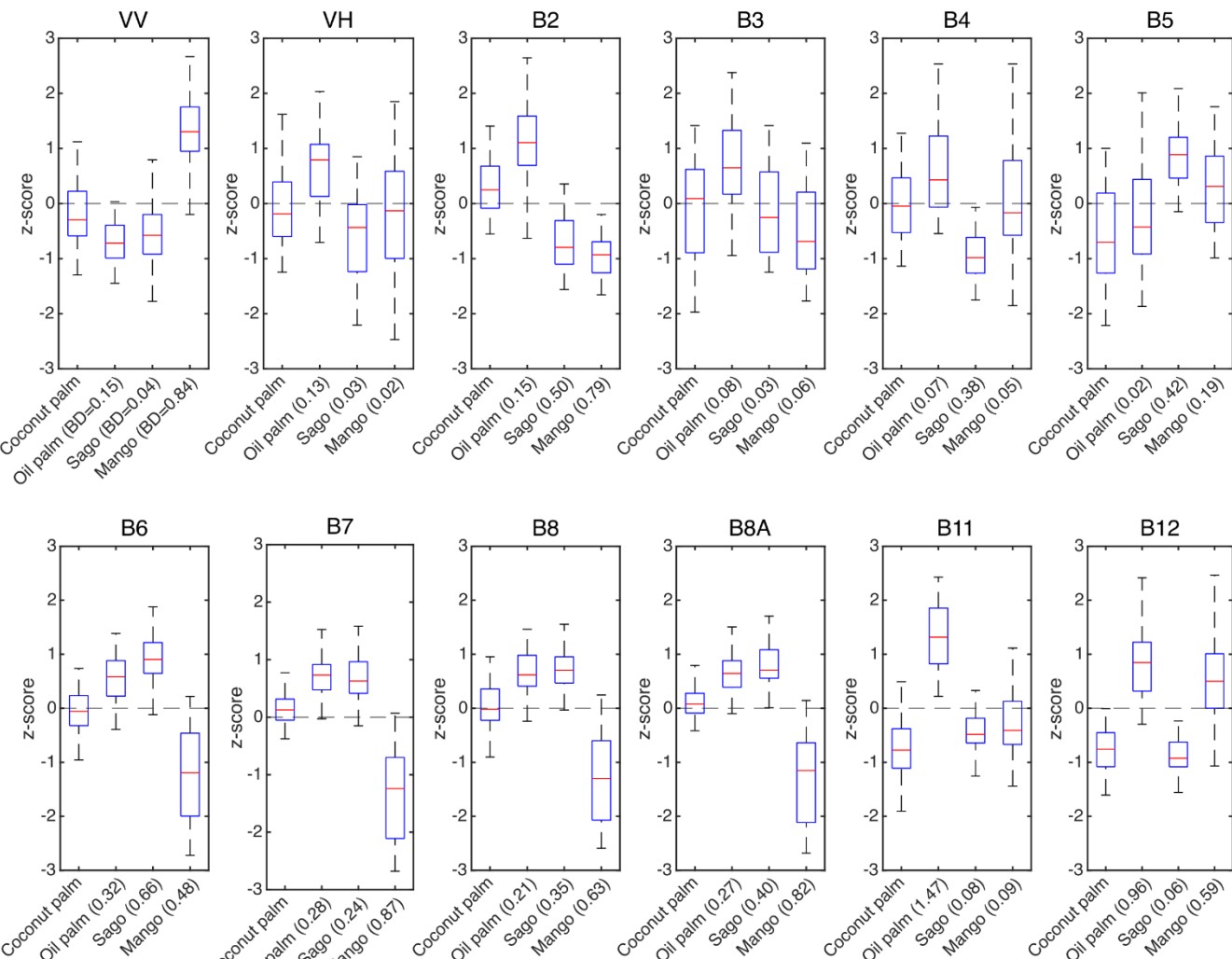

**Figure A2:** Spectral and backscatter separability between coconut palm, oil palm, sago palm, and mango plantations. The overlap between distributions was estimated for the VV and VH bands in Sentinel-1 and for the 10- and 20-meter bands in Sentinel-2. The separability was measured in terms of Bhattacharyya distance (BD) between distributions of coconut palm and other species. The Bhattacharyya distance is displayed in parenthesis in the x-axis. The higher the Bhattacharyya distance the lower the overlap between the two distributions.

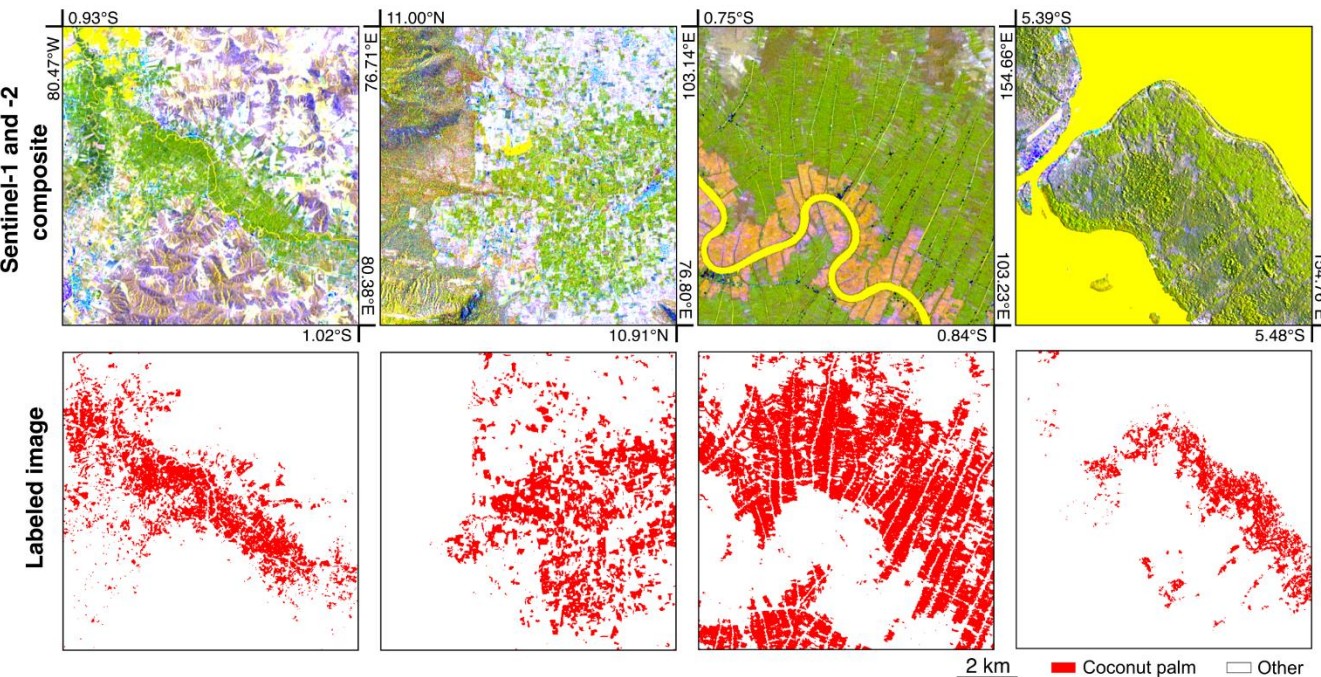

**Figure A3: Example of the 10 x 10 km² images used for training the U-Net model. The training pairs included a Sentinel-1 and Sentinel-2 composite (upper panels) and the corresponding labelled image (bottom panels). The Sentinel-1 and -2 composite includes the polarization bands VV and VH, and the spectral band 11 (short-wave infrared). The labelled image includes two classes: 0 (coconut palm is not present) and 1 (coconut palm is present). The panels show four different coconut-producing regions: from left to right, Manabí Province (Ecuador), Tamil Nadu State (India), Jambi Province (Indonesia), West Kalimantan Province (Indonesia), and Bougainville (Papua New Guinea).**



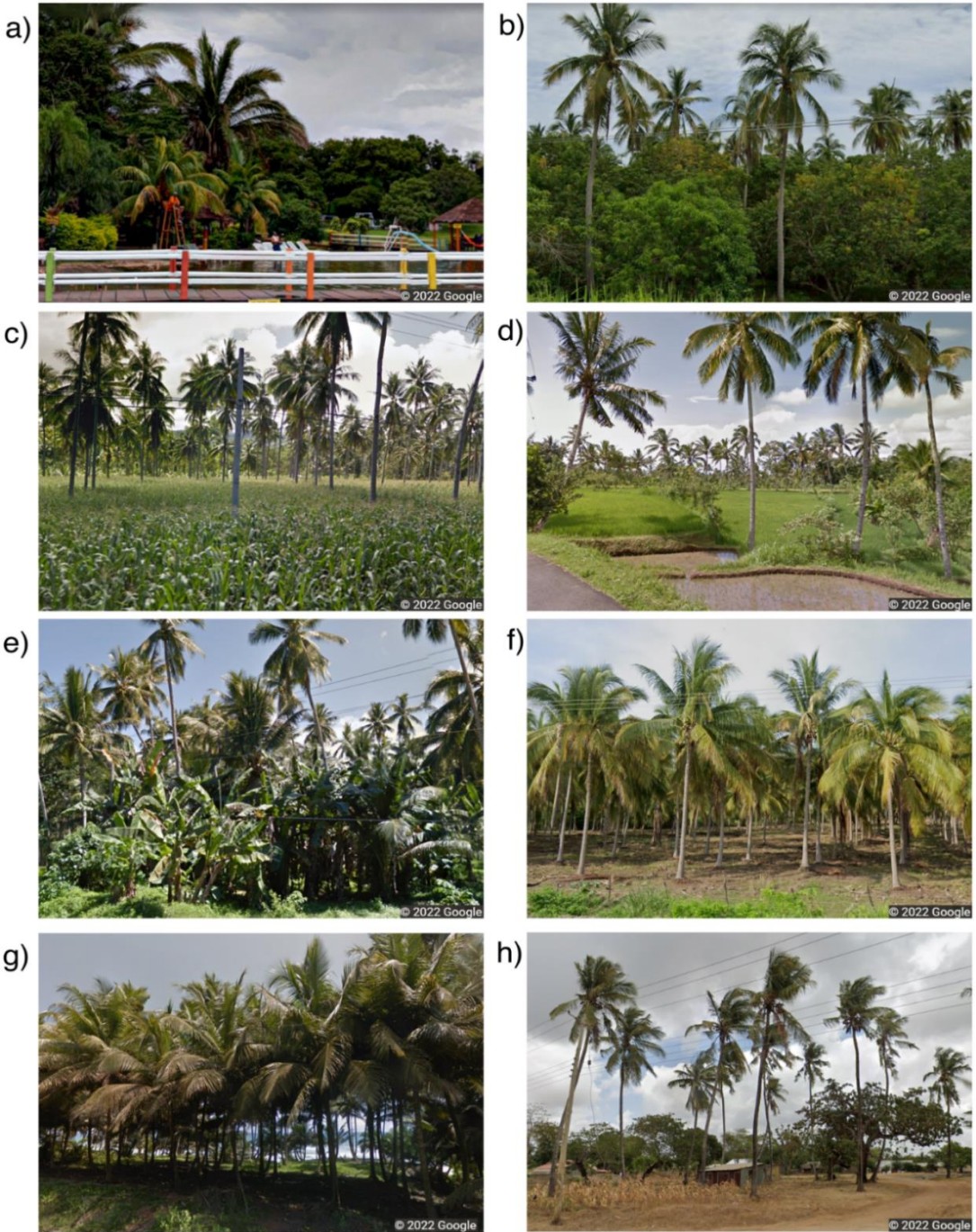

**Figure A4: Images taken from Google Street View @ Google. The images show (a) coconut palms in Bolivia at 808 km from the coast (15.9220°S, 63.1761°W), (b) an intercropping of coconut palm and mango in Mexico (17.2119°N, 100.7382°W), (c) coconut palm and maize in Philippines (5.9776°N, 124.6742°E), (d) coconut palm and rice in Indonesia (8.5596°S, 116.3908°E), and (e) coconut palm and banana in Indonesia (1.0807°S, 103.7871°E), (f) a dense coconut palm plantation in Mexico (18.1230°N, 102.8654°W), (g) dense coastal coconut palm in Indonesia (1.2783°S, 123.5367°E), and (h) sparse coconut palm in Kenya (3.7843°S, 39.8228°E).**

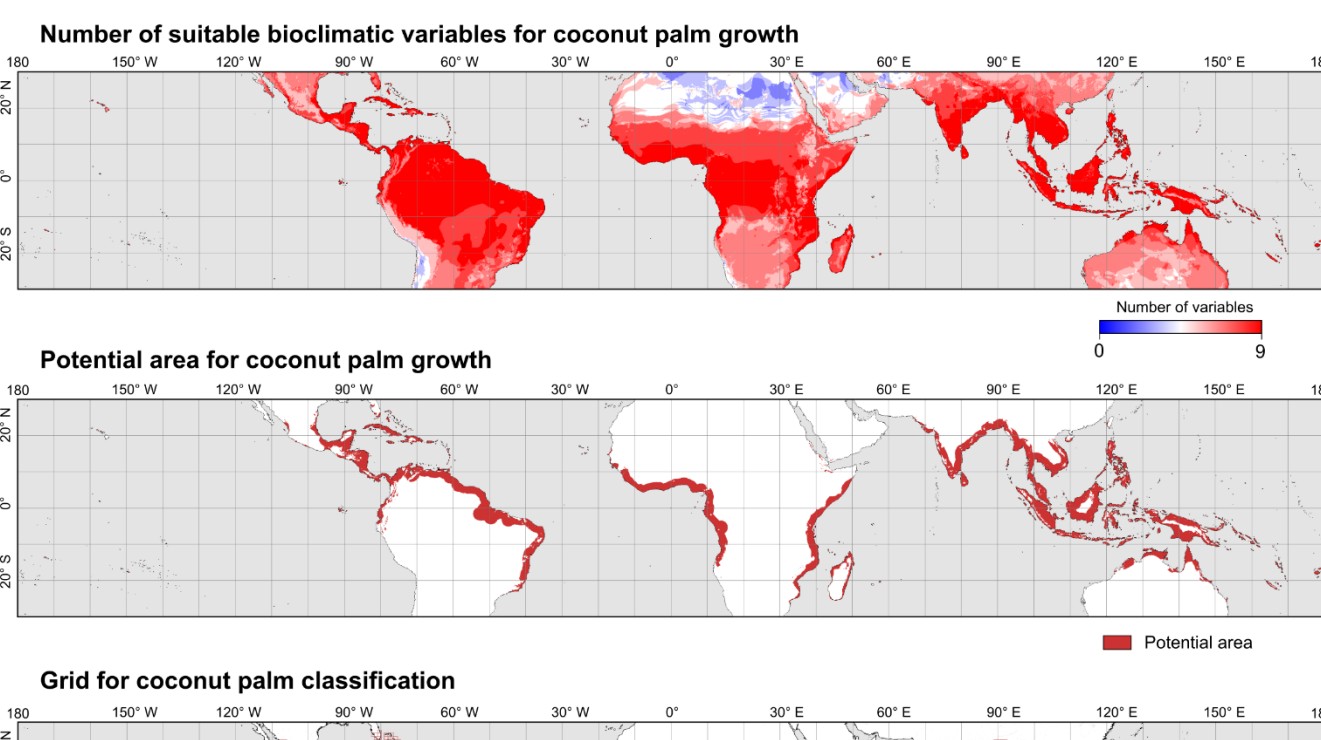

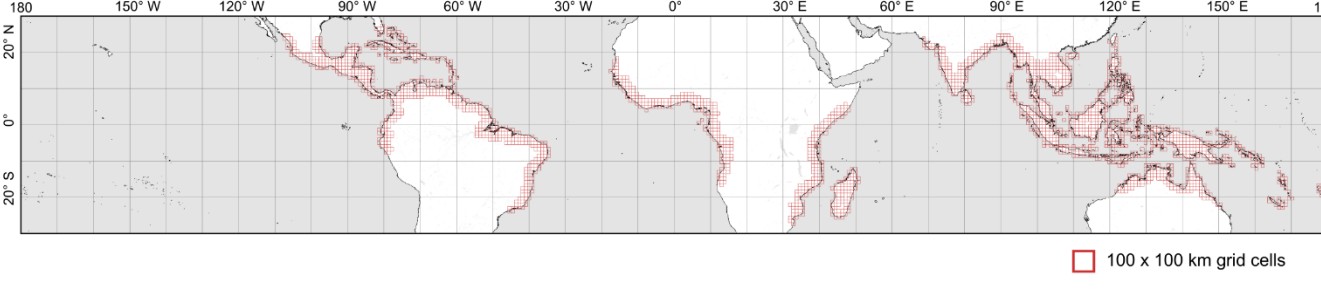

**Figure A5: Maps generated from the bioclimatic analysis. (a) Number of variables that fall within the range of values suitable for coconut palm growth. The bioclimatic variables represent a subset of 8 WorldClim variables and terrain slope that present a low collinearity. The range of values was extracted from 1,139 coconut palm locations. (b) Potential distribution suitable for coconut palm growth. The map represents the pixels with the 9 variables within the range observed in the 1,139 coconut palm locations. Regions inland that are more than 200 kilometres from the coast were masked. (c) 100 x 100 kilometer grid used to classify the Sentinel-1 and Sentinel-2 composites into a land cover map of coconut palm.**

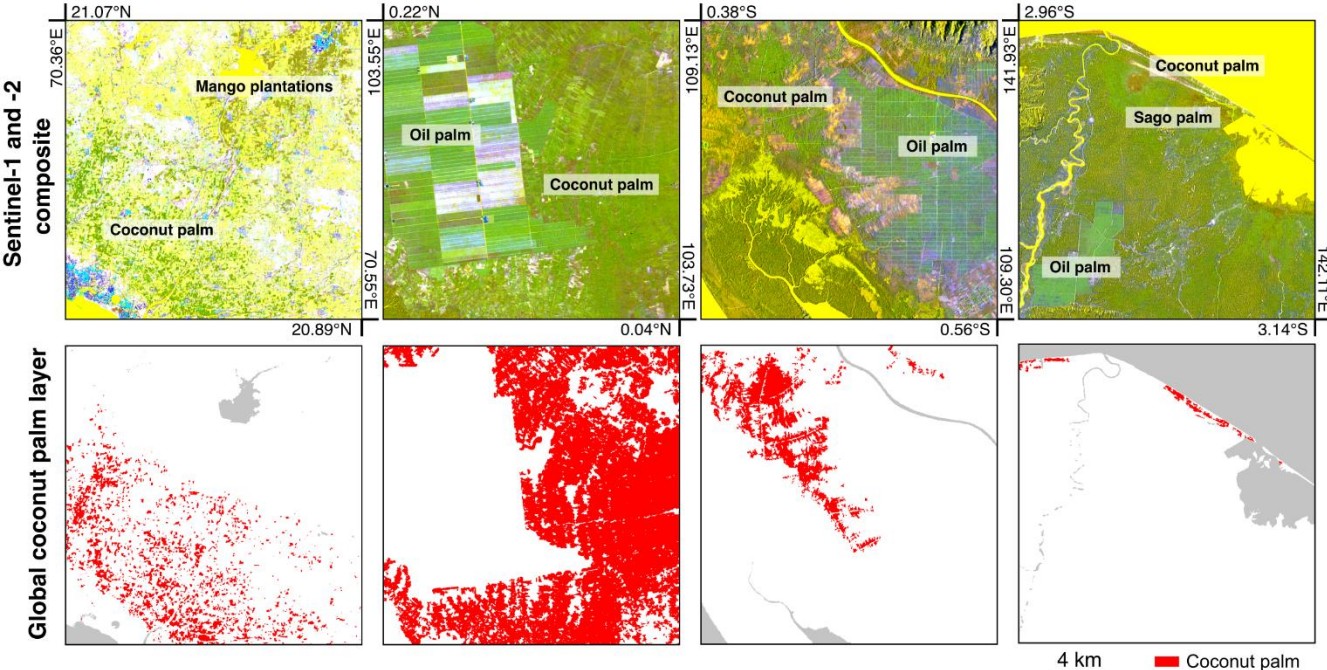

**Figure A6: Classification of Sentinel-1 and Sentinel-2 annual composites into a land cover map of coconut palm.** The Sentinel-1 and -2 composite (upper panels) includes the polarization bands VV and VH, and the spectral band 11 (short-wave infrared). The regions in the panels are, from left to right, Gujarat State (India), Riau Province (Indonesia), West Kalimantan Province (Indonesia), and Sandaun Province (Papua New Guinea). These regions present crops that exhibit similarities to coconut palm in the Sentinel composites. The classification image (bottom panels) shows the global coconut palm layer.

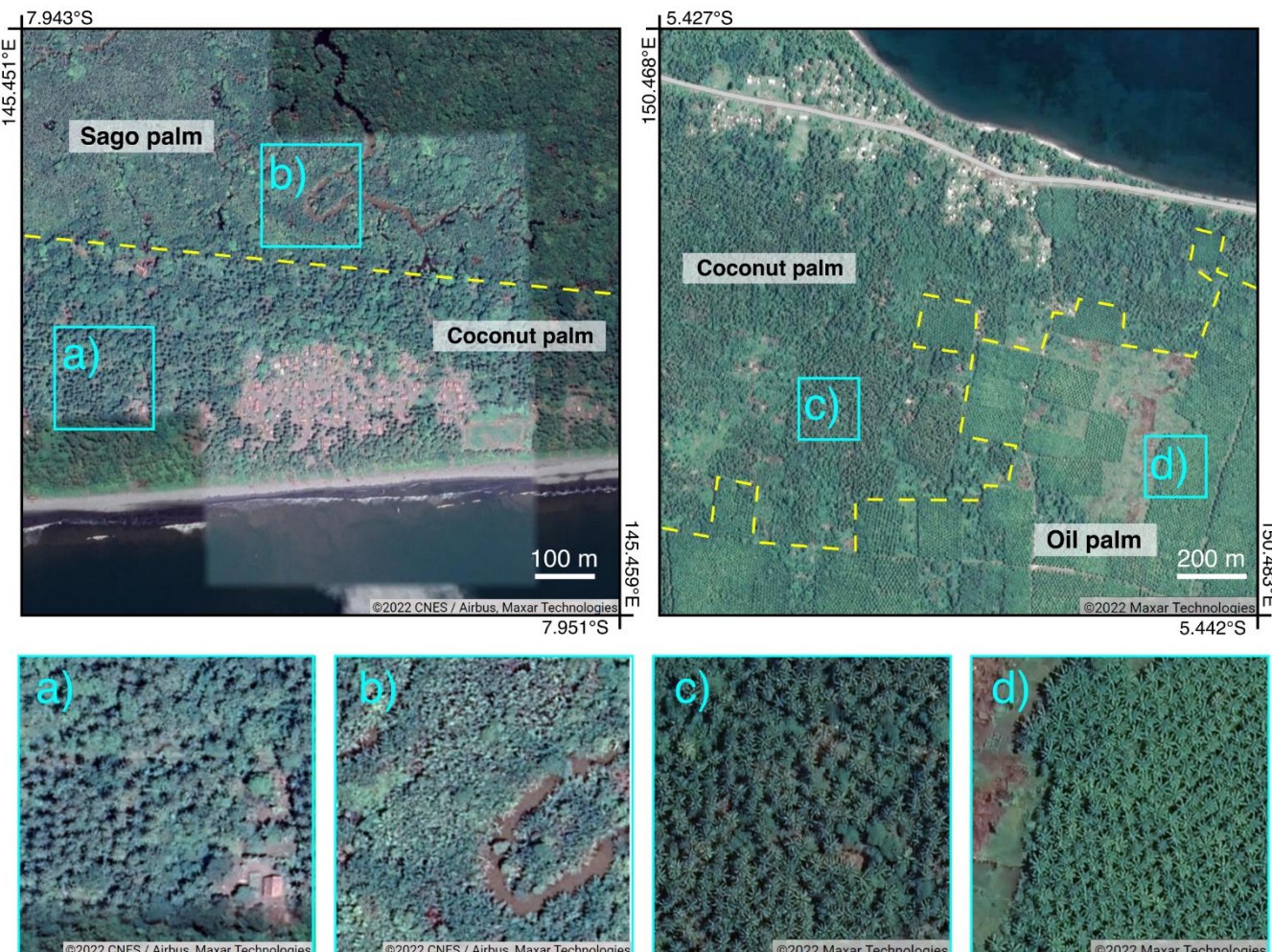


**Figure A7: Sub-meter resolution images in the Gulf (upper-left) and West New Britain (upper-right) Provinces, Papua New Guinea. The images show that coconut palm and other palms (sago and oil palm) grow in separate areas. The bottom panels feature detailed images of coconut palm, sago palm, and oil palm. The satellite images are the sub-meter resolution images that are displayed as the base layer in Google Earth @ Google.**



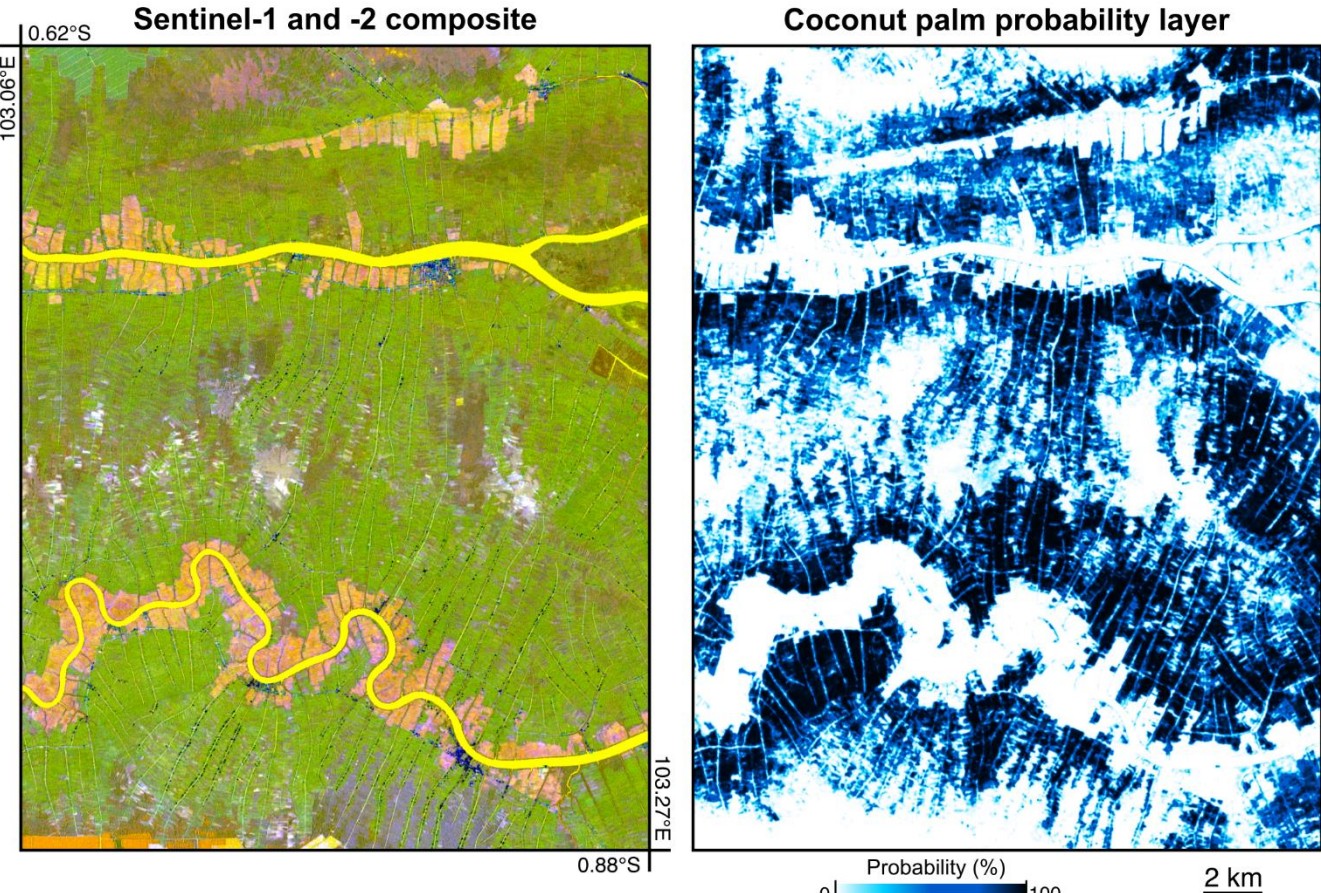

Figure A8: Sentinel-1 and Sentinel-2 annual composite (left panel) and probability layer for the class 'coconut' (right panel) produced with the U-Net model in Riau Province (Indonesia). The Sentinel-1 and -2 composite includes the polarization bands VV and VH, and the spectral band 11 (short-wave infrared). The probability layer represents a score that indicates the confidence level of the classification model in predicting the presence of coconut palm.

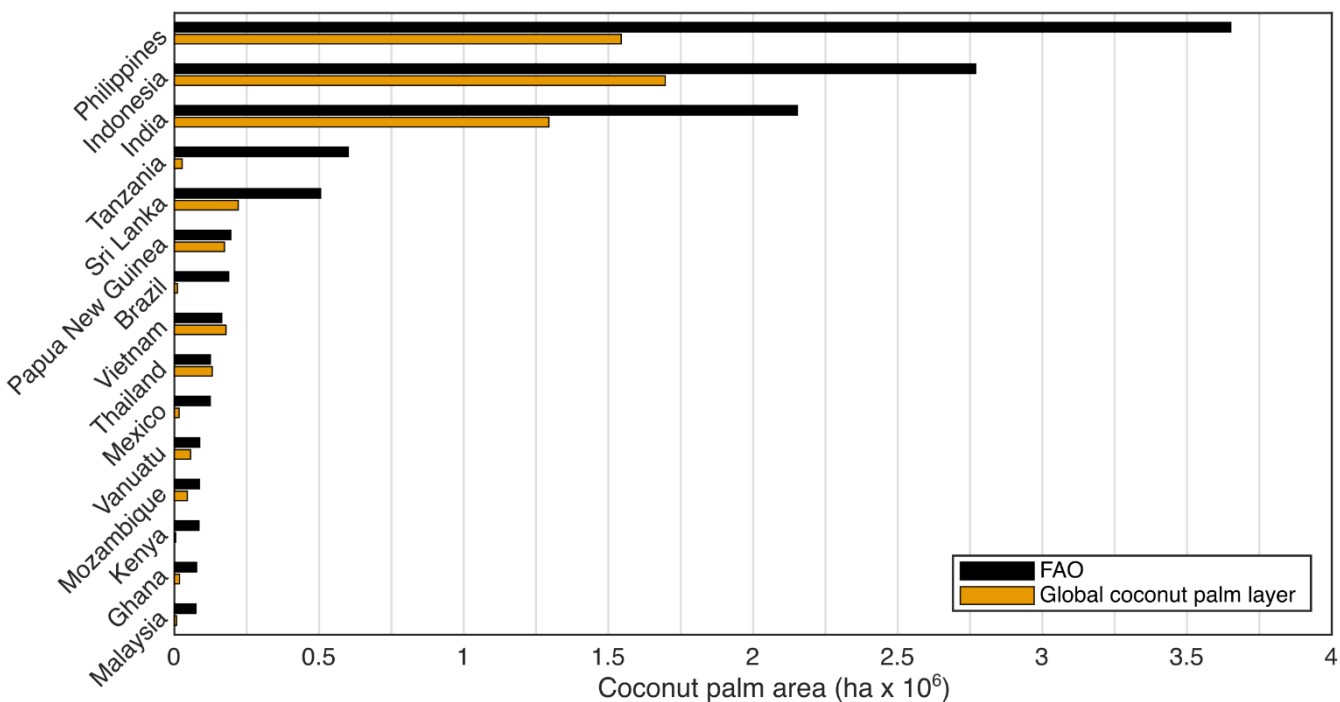

**Figure A9: Coconut palm area mapped using Sentinel-1 and Sentinel-2 and coconut harvested area from FAO for the top 15 coconut-producing countries in 2020.**


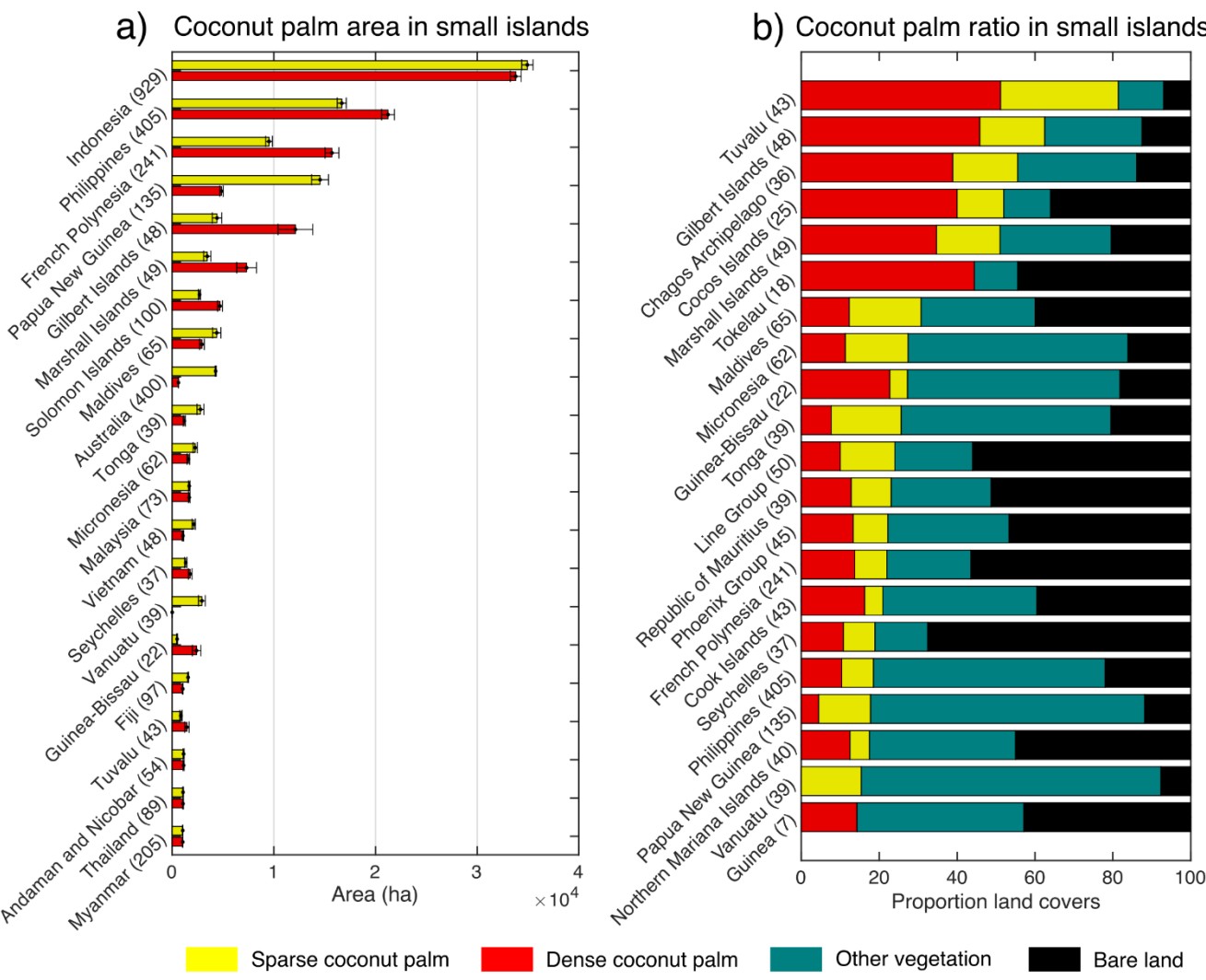

**Figure A10: (a)** Coconut palm area estimates in small tropical islands and **(b)** percentage of the coconut palm area compared to the total island surface per country. The areas were estimated using a sampling-based approach; 5,000 points were randomly sampled in small tropical islands (areas from 1 to 200 ha and between latitudes 30°S and 30°N) and the land cover was identified using sub-meter resolution images. The number between parentheses reflects the number of sampled points in each country. Error bars represent the 95 % confidence interval.