# Peer review of "High-resolution global map of closed-canopy coconut"

_Earth System Science Data, 2022_

## Author Comment (AC1)

**Response to Reviewer #1**

Article entitled "High-resolution global map of closed-canopy coconut", attempt is to produce the first global coconut map at high spatial resolution (20 meters) and estimate the global coconut area using satellite remote sensing. The study results could be a scientific base for the high-resolution, quantitative, and precise data necessary for assessing the relationships between coconut production and the synergies and trade-offs between various sustainable development goal indicators. Therefore, the topic fits the scope of the 'Earth System Science Data' journal.

While reading this paper, limited literature review, lacks logical justifications of contents doesn't lead the reader to the next step. Overall the organization, analysis, methodology is not as developed as I would expect to see such a reputed journal articles. Therefore, the papers truly needs a revision for data and context updation, robust literature review, necessary explanation of each employed methodology as well as results and findings.

However, I have several major concerns about the article, which I listed below:

Response #1: We thank Reviewer #1 for the comments as his/her feedback substantially improved the manuscript. We have thoroughly revised the manuscript and made the necessary corrections. Please see below a detailed response to all the comments.

Abstract

1. Abstract started with vegetable oil crops and their distributional statistics, but paper mainly focuses on coconut distribution. The writing should focus on coconut plantation only.

Response #2: We have removed the mention of vegetable oil crops. The first sentence of the abstract now focuses on coconut palm.

1. Why suddenly deep learning for semantic segmentation, e.g., 'U-Net model' used in this mapping not justified.

Response #3: We have now added a justification of the U-Net model in the Methods section (Lines 151-155):

*"Semantic segmentation is well suited for mapping plantations, such as coconut palms, since it can automatically capture the spatial and contextual information in the image and, as a result, less effort is required compared to feature engineering in standard machine learning (Ma et al., 2019). Such contextual information includes the shape of the plantation, the presence of a harvesting trail, or texture patterns within the plantation."*

We found it informative to mention the type of classification model in the abstract, as it promptly informs the reader about the methodology employed to create the map. However, we have chosen not to include the detailed justification for selecting the U-Net model in the abstract, as we believe that including such details would significantly lengthen the text and potentially diminish the interest of the reader.

1. The study why produced '20-meter global coconut layer' from sentinel 1 and sentinel 2, this is also confusing from 10m sentinel product. It may create confusion to the readers.

Response #4: Sentinel-2 contains bands with different spatial resolutions (10, 20, and 60 meters). Our results have 20-meter resolution because we used a spectral band with 20-meter resolution. We explained this in Lines 145-148:

*"We selected band 11 as the optical band for the classification of coconut palm. Since band 11 has a spatial resolution of 20-meters, we aggregated the Sentinel-1 composites to 20-meters using a bilinear interpolation. As a result, the final coconut palm layer has a spatial resolution of 20 meters."*

We included a section of the methodology dedicated to analyzing the Sentinel-2 bands and justify the selection of band 11 (Section 2.4 *Feature selection*). We consider that these details are not relevant enough for their inclusion in the abstract.

1. Again in the 3rd sentence discussing on 'mapping palm species', reader may lose interest to read for such a confusing mixing between 'coconut' and 'palm species', not clear what authors want to focus.

Response #5: We have now removed this sentence and the abstract focuses only on coconut palm mapping, without any mention of other palm species.

1. Second last sentence should be focus on coconut production instead of 'vegetable oil production'.

Response #6: The abstract now only focuses on coconut palm mapping; in the second last sentence, we have replaced 'vegetable oil production' for 'coconut production', as our coconut palm map also provides the essential data needed for assessing *"the synergies and trade-offs between various sustainable development goal indicators"*.

Introduction

1. The literature review section little poor and required robust literature review.

Response #7: The revised Introduction section now includes references to describe coconut palm distribution and climate suitability (see *Response #8*). Moreover, we have included a new paragraph that provides a comprehensive review on coconut palm mapping and emphasizes the research gap in coconut mapping (Lines 51-60):

*"Sub-meter satellite data and aerial images have been used for detecting individual coconut palms (Zheng et al., 2023; Freudenberg et al., 2019; Zheng et al., 2021), delineating coconut palm canopy (De Souza and Falcão, 2020; Vermote et al., 2020), and coconut palm detection in the context of land cover classification (Burnett et al., 2019). These studies used various methodologies, including threshold-based classification, random forest using feature extraction, and more advanced techniques such as object detection and semantic*

*segmentation using deep learning. Similar efforts have been made to map coconut palm using decametric-scale satellites such as Sentinel-1, Sentinel-2, or Landsat-7 (Lang et al., 2021; Jenifer and Natarajan, 2021; Palaniswami et al., 2006). Another study detected individual coconut palms using airborne laser scanning (Mohan et al., 2019). Despite previous efforts to map coconut palm, these studies have focused on the local and regional scale, and a global coconut palm map has not been produced yet at a high spatial resolution."*

1. Writing should be focused on coconut production and distribution instead focusing on starting with very non-contextual sentence with 'vegetable oil crops' (1st sentence), even discusses '1.7% of the global volume of vegetable oils' (2nd sentence).

Response #8: The Introduction now starts and focuses on describing the coconut palm distribution and climate suitability for coconut production (Lines 32-37).

*"Coconut (Cocos nucifera L.) is a palm species native to tropical islands in the western Pacific but also grows in other tropical areas (Gunn et al., 2011). Climate is an important determinant of coconut palm growth and yield (Peiris and Thattil, 1998). Climate factors such as temperature and relative humidity have been used in descriptive models for predicting coconut yield up to four years in advance (Kumar et al., 2009a). Weather data explained past trends in coconut production (Kumar et al., 2009b), and potential changes in the coconut palm distribution area expected due to climate change in India (Hebbar et al., 2022)."*

1. 3rd sentence "Coconut is generally overlooked ……. see this palm as a threat to biodiversity" is not also confusing, why suddenly palm came in discussion, it bit confusing. Need a few lines of introduction before mixing such terminologies.

Response #9: In the 3rd sentence, 'this palm' referred to coconut palm. The revised manuscript now uses terms such as 'coconut palm', 'sago palm', or 'areca palm' throughout the text, and the term 'palm' is not used alone and ambiguously.

1. Reference started with "Meijaard et al., 2020b" in the first sentence instead of 'Meijaard et al., 2020a', no reference sequence maintained.

Response #10: We have corrected these references. Thanks for noticing this.

2. The sentence "A high resolution global map of the distribution of coconut would serve as a basis for understanding their impacts, as well as help to shape environmental and biodiversity policy". This is not rightly written, is it really possible to "understand their impacts from map", how it can shape 'environmental and biodiversity policy' is also very confusing? This sentence has no value as written with superficial context.

Response #11: We have rephrased and clarified the meaning of this sentence. We also give a brief example on how the map could be used to assess environmental impacts that could potentially provide insights for policy making (Lines 47-49): '*A high-resolution global map of*

*the coconut palm distribution can be used in geospatial analysis to assess environmental impacts and, thus, inform policy (e.g., estimate the extent of coconut plantations in areas of high biodiversity and assess the subsequent impact on biodiversity indices).'*

1. "Research is therefore needed to map coconut on a global scale", are authors map coconut, or 'coconut tree' or 'distribution of coconut tree' or distribution of any species, everything very confusing without any contextual writing and justification from previous sentences.

Response #12: We have now rephrased the sentence to *'Research is therefore needed to map the extent of coconut palms at a global scale'*. Please note that we used the term 'coconut palm' instead of 'coconut tree' throughout the text. Botanically, coconut species does not meet the criteria to qualify as trees since it does not have secondary growth.

1. Last paragraph of introduction "the first global coconut map", is it rightly said 'Coconut map' or something else, please see previous comment to write it appropriately.

Response #13: We have rephrased the sentence to 'the first global coconut palm map' and used the term 'coconut palm' throughout the manuscript. We also use 'global coconut palm map' to consistently refer to the main dataset that results from this study. Thanks for pointing this out.

1. "where climate was favourable for coconut growth" is climate only the factor, as native we know soil is priority concern for coconut growth and their distribution.

Response #14: We have conducted an analysis using a soil map from FAO. The analysis aimed to determine the overlap between black soils and the global coconut palm layer (see results and a detailed response in *Response #29*). Please note that drainage is the most determining soil property for coconut palm growth, and coconut palm can actually grow in a wide range of soil types (Chan and Elevitch, 2006). In the Discussion section (Lines 343-348), we explain this constraint:

*"Coconut palm prefers sandy soils, but different types of soil can support the growth of coconut palm as long as they are well-drained (Chan and Elevitch, 2006), which explains why coconut palm grows in the first few kilometres of coastline in Papua, while sago palm dominates the landscape in inland swampy areas. The drainage requirements for coconut cultivation also explain the unsuitability of vertisols, also known as black soils, which contain a high content of expansive clay minerals with inherent poor drainage. Despite not including a soil map in the bioclimatic analysis, the resulting layer from the coconut palm classification presented a negligible overlap with vertisol areas, for instance the Deccan Traps in India."*

Chan, E., & Elevitch, C. R. (2006). Cocos nucifera (coconut). Species profiles for Pacific Island agroforestry, 2(1), 1-27.

1. Very few literatures addressed in the research gap section, please try to add few more literature in context.

Response #15: We have now included a paragraph to specifically address the research gap regarding the mapping of coconut palm (please see *Response #7*).

Methods:

2.1 Bioclimatic analysis

1. SPAM model suddenly introduced as acronym, without having any discussion.

Response #16: We have now defined SPAM (Spatial Production Allocation Model) in the first mention of the acronym.

2. Title 'bioclimatic analysis' what does it mean? For what purpose this term is used missing here.

Response #17: We have now changed the title of the subsection to 'Bioclimatic analysis for mapping the potential distribution of coconut palm'. Also, we have explained the purpose of this analysis in the first sentence of the subsection (Lines 83-84):

"*We used a bioclimatic analysis to determine the potential coconut-growing regions and, subsequently, constrain the spatial extent of the classification of satellite data.*"

We have also included a new subsection in *Methods* (*Section 2.1. Overview*) which includes a summary of the Methods, provides the logical steps followed to produce the dataset, and provides a guidance for the following subsections of Methods section.

3. Repetition of sentences, example 'a literature search' and 'our literature search'.

Response #18: We have rephrased the sentences to avoid repetition of terms.

4. What does it mean about this sentence - "The points were collected in coconut-producing regions, based on our literature search, and coconut regions according to the SPAM model"?

Response #19: We have now clarified this sentence by providing more details about this procedure (Lines 84-88):

"*[...] we first conducted a literature search to identify regions known for coconut palm cultivation. Additionally, we used the SPAM2010 (Spatial Production Allocation Model) dataset (Yu et al., 2020), which depicts the global occurrence of coconut production across a 5-arcmin grid (Fig. A1). Once the coconut-producing regions were identified, we visualized sub-meter resolution satellite data shown in Google Earth and collected points in locations where coconut palms were present (Fig. 1a).*"

5. About previous sentence mentioned - from secondary data to collect primary location is not a good research, for species distribution model authors need to collect actual location to model it based on bioclimatic variables, soil, or/and other factors.

Response #20: The points depict actual locations of coconut palms. The presence of coconut palms was inferred by visual inspection of sub-meter resolution satellite imagery (see example of this data in Fig. 2) and, when available, images from Google Street View (see Fig. A4). Please also see *Response #24* for more details of the quality assurance of the method and *Response #29* which provides a specific example of how to differentiate between coconut palm and areca palm.

6. "we identified coconut" this is not right, coconut, tree or species, please specify as mentioned earlier.

Response #21: We have specified 'coconut palm' throughout the manuscript.

7. After collecting bioclimatic data, need to know collinearity but here authors failed to understand collinearity before moving to understand bioclimatic responses.

Response #22: We have now included a collinearity diagnosis in the bioclimatic data. We have evaluated the collinearity using the variance inflation factor (VIF). This is explained in the revised manuscript in Lines 95-100:

*" WorldClim V1 Bioclim consists of 19 bioclimatic variables derived from monthly temperature and precipitation. Given that the variables WorldClim were obtained from the same time series, we used the variance inflation factor (VIF) to address collinearity issues. The VIF determines if a set of variables is strongly correlated with each other. A VIF value higher than 5 indicates a high multicollinearity. We removed variables that presented a VIF higher than 5, which resulted in a subset of 8 WorldClim variables and terrain slope (Table A1). We used only the subset of 9 variables in the estimation of the potential coconut palm distribution."*

The results of the new analysis are very similar to the previous version (see below the comparison with and without addressing collinearity). Collinearity issues did not have an impact on the results due to the nature of the methodology. The collinearity would have made a substantial impact if the methodology consisted of a regression model, as done in Hebbar et al., 2022. In that case, addressing collinearity would have been an essential part of the methodology.

[Figure]

Potential area for coconut palm growth in the submitted version (top panel) and in the revised version of the manuscript (bottom panel).

Hebbar, K. B., Abhin, P. S., Sanjo Jose, V., Neethu, P., Santhosh, A., Shil, S., & Prasad, P. V. (2022). Predicting the potential suitable climate for coconut palm (Cocos nucifera L.) cultivation in India under climate change scenarios using the MaxEnt model. Plants, 11(6), 731.

8. Why only bioclimatic, what about soil, geology and geomorphology, which is most important factor of coconut distribution.

Response #23: Climate factors play a key role in coconut palm growth and yield. We have now emphasized this in the Introduction section (Lines 33-37):

*"Climate is an important determinant of coconut palm growth and yield (Peiris and Thattil, 1998). Climate factors such as temperature and relative humidity have been used in descriptive models for predicting coconut yield up to four years in advance (Kumar et al., 2009a). Weather data explained past trends in coconut production (Kumar et al., 2009b), and potential changes in the coconut palm distribution area expected due to climate change in India (Hebbar et al., 2022)."*

In the revised version, we have included terrain slope as a variable representing geomorphology. Results including terrain slope were not substantially different from the previous version (see above figure in *Response #22*). The reason for this is that some regions in the world grow coconut palms on steep terrain. Consequently, the analysis did not consider slope as a critical variable that determines the coconut palm distribution worldwide. Although coconut palms indeed prefer flat terrains, coconut palms are also grown on steep slopes, in particular in the Philippines. In fact, around 60% of coconut palm plantations are in sloping and mountainous areas in the Philippines (Pabuayon et. al, 2008).

In reference to soil types, we have included an analysis to determine the overlap between a black soil map and the global coconut palm layer (see a detailed response in Response #29). We did not include geology (parent rock) since parent rock determines soil type, which is a more direct factor for coconut palm growth.

Please also note that, despite only using climate variables and terrain slope, our bioclimatic analysis produced a potential distribution map that covers the coconut-producing regions of India; see figure below showing coconut palm occurrence points obtained from the Coconut Development Board (CDB), Ministry of Agriculture of India.

[Figure]

Figure taken from Hebbar et al., 2022. The map shows the location of coconut palms obtained from the Coconut Development Board (CDB), Ministry of Agriculture of India.

Last, the sole purpose of the bioclimatic analysis was to constrain the regions where we classified satellite data. Our analysis likely overestimates the potential coconut palm distribution, but this ensured that no coconut palm plantation was omitted in the classification procedure. We have emphasized and mentioned this in different parts of the manuscript, for instance in lines 83-84:

*"We used a bioclimatic analysis to determine the potential coconut-producing regions and, subsequently, constrain the spatial extent of the classification of satellite data."*

Hebbar, K. B., Abhin, P. S., Sanjo Jose, V., Neethu, P., Santhosh, A., Shil, S., & Prasad, P. V. (2022). Predicting the potential suitable climate for coconut (Cocos nucifera L.) cultivation in India under climate change scenarios using the MaxEnt model. Plants, 11(6), 731.

Pabuayon, I. M., Medina, S. M., Medina, C. M., Manohar, E. C., & Villegas, J. I. P. (2008). Economic and environmental concerns in Philippine upland coconut farms: an analysis of policy, farming systems and socio-economic issues. Economy & Environment Program for Southeast Asia, IDRC—CRDI, Singapore.

9. For verification "visualized images from Google Street Maps to verify the presence of coconut", would it be good idea to verify coconut areas? There will be lot of wrong areas which will be looks like other palm or arecanut and others, are not possible to use them to generate coconut distribution area.

Response #24: Coconut palms can be differentiated from other palm species in sub-meter resolution images obtained from Google Maps. However, we acknowledge that the visual identification of coconut palms requires substantial training. To the untrained eye, palm trees look very similar in sub-meter resolution images (Fig. 2). To overcome this challenge, the three data interpreters underwent comprehensive training to ensure the accurate visual recognition of coconut palms in the sub-meter resolution images. In addition, a fourth visual interpreter double-checked the accuracy of the points.

Please note that we did include areca palm in Fig. 2, which shows sub-meter resolution images for different palm species. In the previous version, we used the term 'betel palm' but now consistently use the name 'areca palm' throughout the revised version. Specifically, areca palm can be distinguished given their small canopy compared to coconut palm (please see *Response #27*, which provides several examples of coconut palm and areca palm in sub-meter resolution images).

[Figure]

Figure 2: Sub-meter resolution images depicting (a) coconut palm and (b) other palm species found in the tropics. The images show (from left to right and form up to down) a closed-canopy coconut palm stand in Papua New Guinea (6.124043°S, 134.13848°E) and Indonesia (1.077958°N, 108.966256°E), dense open-canopy coconut palm in Philippines (13.792082°N, 123.016486°E), sparse coconut palm in Kenya (4.367173°S, 39.493028°E), industrial oil palm in Indonesia (1.123642°N, 100.498538°E), semi-wild oil palm in Nigeria (6.641218°N, 5.388639°E), sago in Papua New Guinea (6.122091°S, 134.139178°E), areca palm in India (13.980709°N, 75.632272°E), palmyra palm in Gabon (6.078832°S, 12.330894°E), euterpe palm in Brazil (1.492261°S, 48.3734988°W), attalea palm in Mexico (16.10187°N, 97.396666°W), and Raffia palm in Brazil (4.295997°S, 42.943344°W). The satellite images are the sub-meter resolution images that are displayed as the base layer in Google Earth @ Google.

In Google Street View, areca palm can be identified given its distinctive stem and frond, which are smaller than for coconut palm — see below an image from Google Street View depicting a coconut palm, two areca palms, and oil palm in the background.

[Figure]

Extracted from Google Street View (last accessed: June 21, 2023) https://www.google.com/maps/@1.5485277,101.8903632,3a,42.4y,45.62h,114.22t/data=!3m6!1e1!3m4!1so3W pEpY3T3z8Lq-Vggprcg!2e0!7i16384!8i8192?hl=en

10. "a pixel in the WorldClim dataset was considered suitable for coconut growth if at least 18 of the 19 bioclimatic variables fell within the bioclimatic range", why "18 of the 19" is very confusing, no justification added by authors.

Response #25: We used a threshold of 18 instead of 19 variables in order to produce a less conservative potential distribution map; that is a map that likely overestimates the potential coconut palm area. In this way, we wanted to ensure that no area with coconut palms was omitted in the classification.

Despite this, we agree that the use of 18 variables is a somewhat arbitrary decision. In the revised version of the manuscript, we evaluated the collinearity and reduced the number of variables to 9 non-redundant variables. Then, we set the threshold number of variables to 9 (Lines 102-103):

"[...] a pixel in the WorldClim dataset was considered suitable for coconut palm growth if the 9 selected variables fell within the bioclimatic range."

The potential distribution map using 9 variables was very similar to the one obtained in the previous version (see *Response #22*).

Section 2.2, 2.3, and 2.4, very well written

Response #26: Thank you for the positive feedback.

2.5 Validation

1. "a cost-effective alternative by visually reviewing the sub-meter resolution images from Google Earth because coconut is easily identified using such data", would it be possible to differentiate from other palm or arecanut and others, following figure '2 coconuts sparse' left down figure high density tree is not coconut, coconut open canopy looks like arecanut, how authors validating - this is not convincing method.

Response #27: It is possible to differentiate coconut palms from other palm species using sub-meter resolution data (aka Google Earth images). As mentioned in *Response #24*, the proper identification of coconut palm in sub-meter resolution images requires substantial training. For the specific case mentioned by the Reviewer, areca palm presents a significantly smaller canopy than coconut palm. Also, areca palm is often planted at a much higher density than coconut palms, creating a distinctive texture pattern in the sub-meter satellite images. To demonstrate this, we have identified 6 different locations in India where coconut palm and areca palm plantations grow close to each other. Please see below the sub-meter resolution images and the Google Street View depicting these plantations.

[Figure]

Location 1. Karnataka State. 13°56'33.5"N 75°48'51.9"E

[Figure]

Location 2. Karnataka State. 14°23'21.5"N 75°38'47.8"E

[Figure]

Location 3. Karnataka State. 14°02'18.6"N 75°54'39.0"E

[Figure]

Location 4. Kerala State. 12°33'43.5"N 74°58'02.2"E

[Figure]

Location 5. Karnataka State. 14°21'59.3"N 75°47'50.6"E

[Figure]

[Figure]

[Figure]

Location 6. Karnataka State. 14°32'55.9"N 74°59'37.9"E

We have double-checked the location for sparse coconut palm in Fig. 2, which is the panel that the Reviewer pointed out. The image corresponds to a location near a village in Kenya where coconut palm is grown by smallholders. In that image, the canopy size is too large for areca palm. Moreover, we have confirmed coconut palm plantations in the surroundings of that location using Google Street View.

Result:

1. "The annual temperature ranged between 22.4 and 28.8 °C", is it valid? I know the region very well, the temperature of this region more than 40C now, is it justifiable for valued output.

Response #28: The annual temperature is an aggregate value over the year, not hourly temperature observations, which indeed can reach 40 °C. We have now rephrased: "The mean annual temperature ranged between 22.4 and 28.8 °C". We have now emphasized this in the revised manuscript (Lines 242-244):

*"In the 1,139 points used for the bioclimatic analysis, the mean annual temperature ranged from 22.4 °C (minimum) to 28.8 °C (maximum) (Table A1)."*

Please also note that the range of mean annual temperatures (from 22.4 and 28.8 °C) represents the minimum and the maximum mean annual temperatures extracted from the points collected for the bioclimatic analysis. These points were collected in several regions of the world, not only in India; see Fig. 1a.

1. "Indian state of Karnataka" [black soil] is a decan trap region, coconut production is very less in Karnataka, some of the southern section producing few coconuts, soil associated with geology and geomorphology is most important factor rather distance from and elevation.

Response #29: Following the Reviewer's comment, we have used a black soil map to inspect the spatial overlap between black soils and our product. The black soil map was obtained from FAO. We found very little overlap between our coconut palm layer and the black soil

map; please see figure below. The coconut palm area in black soils only represented the 3.8% of the total coconut palm mapped area in India.

[Figure]

Coconut palm density map (our data) and black soil map in India (FAO).

The black soil map from FAO has a coarse resolution, which very likely explains the overlap with our dataset. We further investigated the overlapping regions and inspected whether we incorrectly detected coconut palms in these regions. We found several coconut palm plantations within the perimeters of the black soil map; please see the figure below showing a coconut palm plantation in Karnataka state in an area depicted as black soil in the FAO layer. Due to the inaccuracies of the black soil map, we discarded its use to mask out the coconut palm layer, since it would result in unwanted omission errors.

[Figure]

[Figure]

Coconut plantations in Karnataka State (14°00'17.0"N, 76°42'19.6"E). According to FAO, the region is supposedly characterized by the presence of black soils.

In summary, as the Reviewer pointed out, black soils are indeed a limiting factor for coconut palm growth. Our classification model was able to detect coconut palm mostly outside the boundaries of black soils. We found large extensions of coconut palm plantations in the south of Karnataka, but these plantations were mostly outside the Deccan Traps; please note that Karnataka State is the second largest coconut palm state in India (see table below) according to the estimates for 2015-2016 from the National Horticulture Board (NHB), an organization under the control of the Ministry of Agriculture and Farmers Welfare of India.

| | STATES | COCONUT (2015-2016) | |
|---|---|---|---|
| | | Area (ha x 10$^3$) | Production (tonnes x 10$^3$) |
| 1 | KERALA | 770.62 | 5113.14 |
| 2 | KARNATAKA | 526.38 | 3529.83 |
| 3 | TAMIL NADU | 459.74 | 4247.12 |
| 4 | ANDHRA PRADESH | 103.95 | 982.42 |
| 5 | OTHERS | 52.22 | 266.90 |
| 6 | ODISHA | 50.91 | 226.00 |
| 7 | WEST BENGAL | 29.51 | 257.11 |
| 8 | MAHARASHTRA | 27.75 | 186.67 |
| 9 | GUJARAT | 22.81 | 215.20 |
| 10 | ASSAM | 19.73 | 91.25 |

Source: Horticulture Crops Estimates for the Year 2015-16 (Final estimates): https://www.nhb.gov.in/ (Last accessed: July 5, 2023)

We thank the Reviewer for the comment. We overlooked the limitations of black soils on coconut production and potential false positives associated with such soils. The analysis above shows that the overlap between our coconut palm map and the black soil map is negligible. We have mentioned this in the Discussion (Lines 341-348):

*"Soil types were not considered in the bioclimatic analysis for the estimation of the potential coconut palm distribution. Coconut palm prefers sandy soils, but different types of soil can support the growth of coconut palm as long as they are well-drained (Chan and Elevitch,*

*2006) (Chan and Elevitch, 2006) [...]. The drainage requirements for coconut cultivation also explain the unsuitability of vertisols, also known as black soils, which contain a high content of expansive clay minerals with inherent poor drainage. Despite not including a soil map in the bioclimatic analysis, the resulting layer from the coconut palm classification presented a negligible overlap with vertisol areas, for instance the Deccan Traps in India."*

1. "the coldest month was 11.5 °C, indicating that coconut cannot tolerate cold temperatures", the regions coldest month temperature is less than 11.5C. I do not think, coconut responsive with climate rather geology and/or soil.

Response #30: We agree with the Reviewer that the text was not accurate, as indeed coconut palms can tolerate hourly temperatures lower than 11.5°C. The temperature 11.5°C is an aggregated value (monthly mean temperature of the coldest month). We have removed the sentence "coconut palms cannot tolerate cold temperatures" and clarified that 11.5°C represents a monthly mean temperature.

1. "Interestingly, we found that coconut grows in a variety of rainfall regimes. Coconut was found in humid tropical regions as well as relatively arid regions in India and East Africa, regions with no precipitation during the driest quarter and where annual precipitation was slightly above 100 mm." This results totally invalid – there is no arid regions with coconut trees, Kerala called 'land of coconut' –average rainfall more than 3000mm. Such huge in-discrepancy results authors will be able to justify, if not include soil and geological condition. In some context local geomorphology also justify some of coconut production like, in the state of Andhra Pradesh.

Response #31: There are arid regions where coconut palm is cultivated, for instance in Dhofar, Oman. However, these plantations use intensive irrigation. We recognize that the text needed clarification, since it gave the notion that coconut palms naturally grow in arid regions. Thanks for noticing this. We have rephrased the text, which now reads (lines 245-250):

*"We found that coconut palm is cultivated in a variety of rainfall regimes. Coconut palm plantations were found in arid and semi-arid regions (annual rainfall <250 mm), such as Dhofar Governorate in Oman (17.0054°N, 54.1069°E), Sindh Province in Pakistan (24.7204°N, 67.5855°E), and Tumbes Province in Peru (4.0481°S, 80.9472°W). However, coconut palm is grown with irrigation in these regions and represents a negligible area compared to the extensive plantations in Kerala State in India, the Philippines, and Indonesia, where rainfall is abundant (annual rainfall >2000 mm)."*

1. "The separability analysis revealed a low separability between coconut and oil palm in the VV and VH bands", as paper focusing coconut, suggest to mention coconut only.

Response #32: We agree that this analysis should not be the central focus of the study, as the journal primarily aims to present datasets. Therefore, we have condensed the explanation of this analysis and move it to the Methods section, as this analysis focuses on the methodology and not on the dataset. We moved the corresponding figure to the appendices.

We kept the analysis and figure (in the appendices) because it justifies the choice of satellite bands used in the classification model.

Following the Reviewer's suggestion, we have also removed the paragraph in the Introduction section regarding the distinctive backscatter values in oil palm. The revised text only mentions that the differences between coconut palm and oil palm are unclear in satellite remote sensing, which makes coconut palm mapping a challenging task.

1. Paragraph 4 in results section, discuss again tree species focusing non-palm plantations only on cinnamon (Cinnamomum spp. Schaeff.) and mango. (Mangifera spp. L.), which is very rare species in the states with coconut tree available in India. These section can be avoided, as never mentioned earlier any species classification earlier throughout the manuscript.

Response #33: The dataset provided in the study has global coverage. There are regions in the world where coconut palm is grown alongside cinnamon and mango. More specifically, coconut palm and mango trees are grown in the Pacific coast of Mexico. Also in India, coconut palm and mango trees are grown in Gujarat State; coconut palm is grown between the cities of Mangrol and Veraval, while mango is grown a few kilometers inland from this location.

Following the Reviewer's comment, we included mango and other tree crop in the revised Methods section (Lines 130-135):

*"[...] there were coconut palm plantations incorrectly classified as oil palm in the global oil palm layer (Descals et al., 2021), indicating that a spectral band other than band 4 could better distinguish oil palm from coconut palm. We also found in our preliminary analysis that sago forests (Metroxylon sagu Rottb.) and mango plantations (Mangifera spp. L.) could also be confused with coconut palm in the VV-VH-band 4 composites. Thus, we inspected the spectral separability between coconut palm, oil palm, sago palm, and mango plantations for all 10- and 20-meter Sentinel-2 bands."*

As a consequence, we have extended the spectral separability analysis to other tree crops that were problematic in the classification. This analysis, however, is briefly mentioned in the main manuscript following the suggestion from a previous comment (*Response #32*).

1. "the global coconut area was 12.31 ± 3.83 x 106 ha", if we see map 4 –estimated area will be higher, although in map now showing some prominent coconut areas in Thailand in North Pattaya region, higher in southern west Bengal, where very few coconut trees available. Not a single plantation available in west Bengal in reality. Lot of miss-classified species included in classification.

Response #34: The Reviewer might be referring to the west of Bangkok city instead of North Pattaya city. Coconut palm plantations in North Pattaya city were negligible in our map. The region close to Bangkok city, where we indeed detected a prominent coconut palm area, includes the Samut Songkhram province and corresponds to one of the largest coconut hotspots of the country. According to the Office of Agricultural Economics (see reference at

the end of this response), Samut Songkhram province is the 5$^{th}$ largest province (out of 77 provinces) in terms of coconut plantation area. Coconut plantations covered 57,175 rai (= 9,148 ha) in the Samut Songkhram province in 2020. We did not make any changes to the layer for this region, as coconut plantations were correctly mapped.

About West Bengal, the state does grow coconut palm but, indeed, the total area in that state only represents the 3.8% of total coconut palm in India (see table in *Response #29*). Following the Reviewer's comment, we have thoroughly inspected the layer and improved it for West Bengal. We found that the classification model was able to detect sparse coconut palms at a higher rate than other regions of the world, explaining the overestimation in West Bengal. Since our map focuses on closed-canopy coconut palms, we have masked out the high density of pixels detected as coconut palm in the inner part of West Bengal. The new manuscript includes a new version of the coconut palm layer, which incorporates the these improvements.

Last, the previous figure gave the incorrect notion that our layer overestimated coconut palm worldwide and in India, in particular, as noticed by the Reviewer. This issue relates to the presentation of the results rather than the accuracy of the dataset. We have removed Fig. 4 and Fig. 5 and have provided two new figures that better represent our results. The new Fig. 4 shows the 100 x 100 km grid cells where coconut palm was detected using the U-Net model. The figure provides a broad overview of the coconut palm occurrence. In addition, the new Fig. 5 shows the density map in India and SE Asia, where most of the coconut palm plantations are found. The density map represents an improvement compared to the previous Fig. 4, as it better depicts the coconut hotspot regions and doesn't give the misleading impression that our layer overestimates the coconut palm area.

Please note that we employed a scientifically rigorous method to assess the accuracy of the layer. We followed the established guidelines outlined in the highly cited paper by Olofsson et al., 2015, which offers a sampling-based approach for evaluating the accuracy of land cover maps. In contrast to the visual inspection of the map figures, the accuracy metrics provide a quantitative measurement of the validity of the layer. Based on the results of the accuracy assessment, we discussed the pros and cons of our layer, being the major drawbacks of our product (a) the high omission of coconut palms in low-density setting and (b) residual false positives in other palm species.

**Grid cells with detected coconut palm using the U-Net model**

[Figure]

Grid cells with detected coconut palm    ▬ No data

Figure 4: Global occurrence map of coconut palm. Grid cells in red depict areas where closed-canopy coconut palm was detected using a U-Net model and annual Sentinel-1 and Sentinel-2 composites for 2020. The cell size is 100 x 100 km. Dark grey represents areas where Sentinel-1 or Sentinel-2 were not available.

**Global coconut palm layer — Density of coconut palm**

[Figure]

Figure 5: Density of coconut palm in India and Southeast Asia. The map was generated from the 20-m global coconut palm layer. The density map highlights the primary areas of coconut production in the region.

Office of Agricultural Economics (Ministry of Agriculture and Cooperatives of Thailand): Coconut production Mature coconut: perennial area Fruiting area, yield and productivity per fruiting area, by province, year 2020. https://www.oae.go.th/assets/portals/1/fileups/prcaidata/files/Coconut%2063(2).pdf (Last accessed 5 July 2023). The document is not available in English. Samut Songkhram province translates to สมุทรสงคราม in Thai, and the second column depicts plantation area in rai (ไร่) units.

Olofsson, P., Foody, G. M., Herold, M., Stehman, S. V., Woodcock, C. E., & Wulder, M. A. (2014). Good practices for estimating area and assessing accuracy of land change. Remote sensing of Environment, 148, 42-57.

Discussion: this section very well written

Response #35: Thank you for the positive feedback on the Discussion section.

Conclusion:

1. "Our global coconut layer is of considerable interest to researchers…. of vegetable oil crops (especially oil palm) to meet future demands for food, feed, biofuel, surfactants, and other oil uses" – focusing on coconut would be better in conclusion again avoiding 'vegetable oil crops'.

Response #36: We have removed the sentence and the conclusion now only focuses on coconut palm.

1. Last sentence "coconut presents a spatial overlap with high levels of threatened species, species endemism, and species richness, in tropical islands and, thus, the global coconut layer might benefit studies that evaluate the associated environmental impacts of coconuts in such biodiversity hotspots"- should be avoided, this is non-contextual from the present analysis.

Response #37: We agree. The spatial overlap between coconut palm extent and biodiversity was not derived directly from the results. We have now rephrased this sentence and mention the analysis as a future study (Lines 456-458): *'the global coconut palm layer can be used in geospatial analysis to assess the spatial overlap between coconut palm extent and areas of highly threatened species, species endemism, and species richness on tropical islands. In this regard, the coconut palm map presented in this study can be valuable for studying the environmental impacts associated with coconut cultivation in biodiversity hotspots.'*

Appendix:

1. Analyzed variables for coconut distribution is not sufficient, need to include many indicators like geology, soil, geomorphology, else Indian coconut distribution will not be justified.

Response #38: Please see *Responses #23 and #29* regarding the need for including other variables such as geology, soil, and geomorphology. In summary, we have included the terrain slope in the analysis as an indicator of geomorphology. We have also used a map of black soils to inspect false positives in regions such as the Deccan Traps in India. However, we did not include the soil map in the analysis; including such coarse-resolution maps would potentially result in omission of areas where coconut palm is grown.

2. Figure A1, 'no of bioclimatic variable' and maps not meaning anything what authors want to mean.

Response #39: We have now changed the title of Figure A1 (now Figure A4 in the revised manuscript) to make it as self-explanatory as possible: *"Number of suitable bioclimatic variables for coconut palm growth"*. The description of the panel has been rephrased as well: *"Number of variables that fall within the range of values suitable for coconut palm growth. The bioclimatic variables represent a subset of 8 WorldClim variables and terrain slope (9 variables) that present a low collinearity."*. Thanks for noticing this.

---

## Author Comment (AC2)

**Response to Reviewer #2**

This research developed the first 20-meter global coconut maps using U-Net and Sentinel imagery. According to the results shown in the paper, the dataset can provide essential and detailed spatial distribution for global coconuts, and the accuracy is relatively high. However, it is recommended to address the mentioned points to improve the clarity, comprehensiveness, and applicability of the research.

Response: Thank you for the positive assessment of the interest in the dataset. We have now addressed your insightful comments. By doing so, the article has substantially improved. Thanks.

1. Please summarize the previous efforts and methods developed for coconut tree monitoring in the introduction section. This will help readers understand the novelty and significance of the current research.

Response: We have now included a paragraph that summarizes previous studies on coconut palm mapping and explains why our study is novel by saying that there is no global coconut palm map with such level of spatial detail (Lines 51-60):

*"Sub-meter satellite data and aerial images have been used for detecting individual coconut palms (Zheng et al., 2023; Freudenberg et al., 2019; Zheng et al., 2021), delineating coconut palm canopy (De Souza and Falcão, 2020; Vermote et al., 2020), and coconut palm detection in the context of land cover classification (Burnett et al., 2019). These studies used various methodologies, including threshold-based classification, random forest using feature extraction, and more advanced techniques such as object detection and semantic segmentation using deep learning. Similar efforts have been made to map coconut palm using decametric-scale satellites such as Sentinel-1, Sentinel-2, or Landsat-7 (Lang et al., 2021; Jenifer and Natarajan, 2021; Palaniswami et al., 2006). Another study detected individual coconut palms using airborne laser scanning (Mohan et al., 2019). Despite previous efforts to map coconut palm, these studies have focused on the local and regional scale, and a global coconut palm map has not been produced yet at a high spatial resolution."*

2. Provide details of the training samples used in this study. Please present an example of one image (Sentinel-1 and -2 composite) and one corresponding per-pixel label image to illustrate the data used for training.

Response: We have now added Fig. A3, which shows four paired Sentinel composites and per-pixel label images for different coconut palm regions.

[Figure]

Figure A3: Example of the 10 x 10 km$^2$ images used for training the U-Net model. The training pairs included a Sentinel-1 and Sentinel-2 composite (upper panels) and the corresponding labelled image (bottom panels). The Sentinel-1 and -2 composite includes the polarization bands VV and VH, and the spectral band 11 (short-wave infrared). The classification image includes two classes: 0 (coconut palms are not present) and 1 (coconut palms are present). The panels show four different coconut-growing regions: from left to right, Manabí province (Ecuador), Tamil Nadu state (India), Jambi province (Indonesia), West Kalimantan province (Indonesia), and Bougainville (Papua New Guinea).

We included details about the training dataset in Lines 156-164:

*"Semantic segmentation models require image data with a fixed size for both training and prediction. We set the size of the input images to 512 × 512 pixels, which is approximately 10 × 10 km in a 20-meter resolution image. The collection of training data consisted of digitizing polygons in regions that were identified in the bioclimatic analysis. The polygons were drawn in 146 training images (Fig. 1b) using sub-meter resolution to discriminate coconut palm plantations from other land covers. The sub-meter resolution images were the images displayed as the base layer in Google Earth. The U-Net was used for binary classification of coconut palm (digitized polygons) and the rest of land covers (image background; see Fig. A3) and, thus, the resulting layer was a binary raster, in which each pixel presented values of 0 (coconut palms are not present) and 1 (coconut palms are present)."*

3. The feature differences between different crop trees (such as coconut, oil palm, mango as mentioned in the paper) should be given for better comprehension of the classification procedure.

Response: We have now extended the spectral separability analysis to the other crop trees mentioned in the main text (sago and mango plantations), which were the most problematic plantations after oil palm. We have updated Fig. A2 to include these other crop trees.

[Figure]

Figure A2: Spectral and backscatter separability between coconut palm and oil palm, sago, and mango plantations. The overlap between distributions was estimated for the VV and VH bands in Sentinel-1 and for the 10- and 20-meter bands in Sentinel-2. The separability was measured in terms of Bhattacharyya distance (BD) between distributions of coconut palm and oil palm points. The Bhattacharyya distance is displayed in parenthesis in the x-axis. The higher the Bhattacharyya distance the lower the overlap between the two distributions.

4. Is it possible to investigate the correlation between the mapping results and FAO statistics on a country level (besides the three countries shown in Table 2)? This analysis will strengthen the validity and reliability of the mapping results.

Response: The manuscript now includes Fig. A9, which depicts the FAO statistics and the coconut palm mapped area on a country level. We have now used this figure to emphasize that our map depicts closed-canopy coconut palm, while sparse coconut palm is mostly omitted. The high rate of omission in sparse coconut palm likely explains the gap between FAO statistics and our product:

[Figure]

Fig. A9: Coconut palm area mapped using Sentinel-1 and Sentinel-2 and coconut palm harvested area from FAO for the top 15 coconut-producing countries in 2020.

5. Compare different semantic segmentation models or explain why U-Net was chosen as the final model for mapping. Additionally, discuss the potential use of other deep learning models, and elaborate on why semantic segmentation models were preferred.

Response: We have included the justification for using a semantic segmentation model (U-Net model) in Lines 152-153:

"*Semantic segmentation is well suited for mapping plantations, such as coconut palm, since it can automatically capture the spatial and contextual information in the image and, as a result, less effort is required compared to feature engineering in standard machine learning (Ma et al., 2019). Such contextual information includes the shape of the plantation or texture patterns within the plantation.*"

Also, we have elaborated on the potential use of other deep learning techniques and explained why semantic segmentation was preferred in our study (Lines 329-335 in the Discussion section):

"*Object detection using deep learning applied to very high-resolution images (<1 meter), such as those obtained by DigitalGlobe or Planet, offers great potential for the detection of individual coconut palms (De Souza and Falcão, 2020; Vermote et al., 2020; Freudenberg et al., 2019). This approach could be used to detect coconut palm plantations with incomplete canopy closure and coconut palms that are scattered across the land. In our study, the decametric resolution of Sentinel-1 and Sentinel-2 images made the use of object detection techniques unfeasible. Object detection using deep learning and sub-meter images could*"

*complement our closed-canopy coconut palm layer and could also be useful for mapping different palm trees, including coconut palm, oil palm, and sago palm."*

We have not included a comparison of semantic segmentation models. Following the recommendations in *The aims & scope* of the journal, '*any comparison to other methods is beyond the scope of regular articles.*'

6. Incorporate the coconut-producing regions obtained based on the literature review and SPAM. It might be useful to overlay these regions with the coconut locations shown in Figure 1a to demonstrate their spatial distribution.

Response: The manuscript now includes Fig. A1, which depicts the coconut palm layer in the SPAM dataset. Most of the coconut-producing regions obtained in the literature review depict general administrative regions (districts, provinces, states, etc.). We consider that a map showing these administrative boundaries would not be very informative for the reader. Instead of displaying the administrative boundaries, we believe that the points extracted from these regions (illustrated in Fig. 1a) are more informative and meaningful.

We opted to display the SPAM layer and the point dataset (Fig. 1a) in separate figures. Overlaying one on top of the other would obscure either layer.

[Figure]

Figure A1: Coconut palm map extracted from the Spatial Production Allocation Model for 2010 (SPAM2010).This layer represents areas where the extent of coconut palm plantations exceeds 50 hectares within each ~ 9 km$^2$ pixel off the SMAP dataset.

7. As this study took the semantic segmentation model as the mapping network, it may be helpful to provide pixel-level validation results in addition to the accuracy obtained from validation points, as it will provide a more comprehensive evaluation of the model's performance and its ability to accurately delineate coconut tree boundaries.

Response: We have now included the probability layer as an auxiliary data to the global coconut palm layer. We have explained the probability layer in the Methods section (Lines 164-166):

*"In addition, we generated a probability layer using the second-last layers of the convolutional neural network. Rather than probability layers, the second-last layers represent a confidence score (ranging from 0 to 100) for each class prediction. The probability layer we provide corresponds to the second-last layer of the class 'coconut'."*

We have also added a new figure (Fig. A8) showcasing the probability layer.

[Figure]

Figure A8: Sentinel-1 and Sentinel-2 annual composite (left panel) and probability layer (right panel) produced with the U-Net model in Riau province (Indonesia). The Sentinel-1 and -2 composite includes the polarization bands VV and VH, and the spectral band 11 (short-wave infrared). The probability layer represents a score that indicates the confidence level of the classification model in predicting the presence of coconut palm.

8.  Is it possible to classify the coconut to Sparse, Dense open-canopy and Closed-canopy? It will provide more detailed mapping results, and give the readers opportunity to choose the product they need, just like the previous global oil palm product the authors developed, which provide the smallholder oil palms and industrial oil palm plantations.

Response: Unfortunately, it is not possible to produce a comprehensive map of sparse, dense open-canopy and closed-canopy with the methodology presented in this study and using Sentinel-1 and Sentinel-2. We have mentioned, however, that the potential use of the probability layer could be used as a proxy for coconut palm density (Lines 316-317):

*"We also generated a probability layer that provides a score indicating the confidence level of the model output. This probability layer could serve as a proxy for coconut palm density."*

Please note that a similar issue occurred for the global oil palm plantation in our previous study (Descals et al., 2019). We successfully trained a model that distinguished between smallholders and industrial oil palm plantations. However, the model's output was accurate only if the plantation presented a closed canopy.

Descals, A., Wich, S., Meijaard, E., Gaveau, D. L., Peedell, S., & Szantoi, Z. (2021). High-resolution global map of smallholder and industrial closed-canopy oil palm plantations. Earth System Science Data, 13(3), 1211-1231.

9. Include zoomed-in regions that contain both coconut trees and oil palms (or other plantations) to illustrate the model's capability to accurately extract coconuts from other tree types. This may help to address the statement made by the authors: "The separability analysis revealed a low separability between coconut and oil palm in the VV and VH bands."

Response: We have now included Fig. A6, which illustrates zoomed-in regions that grow coconut palm and other plantations (oil palm, mango, and sago). Thanks for the suggestion.

[Figure]

Figure A6: Classification of Sentinel-1 and Sentinel-2 annual composites in regions with the presence of other crops that exhibit similarities to coconut palm in the Sentinel composites. The Sentinel-1 and -2 composite (upper panels) includes the polarization bands VV and VH, and the spectral band 11 (short-wave infrared). The regions are, from left to right, Gujarat state (India), Riau province (Indonesia), West Kalimantan province (Indonesia), and Sandaun Province (Papua New Guinea). The classification image (bottom panels) shows the coconut palm plantations in red.

10. Provide the uncertainty map of the classification results to help users better understand and utilize the dataset shared in this research.

Response: The probability layer has been now included as an auxiliary data to the classification layer. The layer was uploaded to Google Earth Engine to facilitate user access. We have included a figure showing the probability layer and have included an explanation of this new layer (see response to point 7). Thanks for the suggestion. We hope the probability layer enables users to have a better understanding and use of the dataset.